# Multiplex-GAM: genome-wide identification of chromatin contacts yields insights overlooked by Hi-C

Robert A. Beagrie[1,2,3,11], Christoph J. Thieme [1,11], Carlo Annunziatella[4,11], Catherine Baugher[5], Yingnan Zhang[5], Markus Schueler[1], Alexander Kukalev[1], Rieke Kempfer[1,6], Andrea M. Chiariello [4], Simona Bianco [1,4], Yichao Li [5], Trenton Davis[5], Antonio Scialdone [7,8,9], Lonnie R. Welch [5,12] ✉, Mario Nicodemi [1,4,10,12] ✉ & Ana Pombo [1,6,12] ✉

Technology for measuring 3D genome topology is increasingly important for studying gene regulation, for genome assembly and for mapping of genome rearrangements. Hi-C and other ligation-based methods have become routine but have specific biases. Here, we develop multiplex-GAM, a faster and more affordable version of genome architecture mapping (GAM), a ligation-free technique that maps chromatin contacts genome-wide. We perform a detailed comparison of multiplex-GAM and Hi-C using mouse embryonic stem cells. When examining the strongest contacts detected by either method, we find that only one-third of these are shared. The strongest contacts specifically found in GAM often involve 'active' regions, including many transcribed genes and super-enhancers, whereas in Hi-C they more often contain 'inactive' regions. Our work shows that active genomic regions are involved in extensive complex contacts that are currently underestimated in ligation-based approaches, and highlights the need for orthogonal advances in genome-wide contact mapping technologies.

Our understanding of gene regulation has been dramatically transformed by genome-wide methods for identifying regulatory elements (for example ChIP-seq, ATAC-seq)[1] and by technologies that show how these elements are connected to one another through 3D genome conformation (for example, Hi-C)[2]. However, many cell types of interest are too rare to assay using these methods. Although single-cell variants of Hi-C are available, they require purified, disaggregated cell suspensions, which can be unachievable for rare cell types embedded in complex tissues. Furthermore, methods based on chromatin conformation capture usually focus on contacts between pairs of elements, neglecting higher-order associations. We previously showed that genome architecture mapping (GAM) can identify three-way chromatin contacts and

[1]Max-Delbrück-Center for Molecular Medicine in the Helmholtz Association (MDC), Berlin Institute for Medical Systems Biology (BIMSB), Epigenetic Regulation and Chromatin Architecture Group, Berlin, Germany. [2]Laboratory of Gene Regulation, Weatherall Institute of Molecular Medicine, Oxford, UK. [3]Chromatin and Disease Group, Wellcome Centre for Human Genetics, Oxford, UK. [4]Dipartimento di Fisica, Università di Napoli Federico II, and INFN Napoli, CNR-SPIN, Complesso Universitario di Monte Sant'Angelo, Naples, Italy. [5]School of Electrical Engineering and Computer Science, Ohio University, Athens, OH, USA. [6]Humboldt-Universität zu Berlin, Berlin, Germany. [7]Institute of Epigenetics and Stem Cells, Helmholtz Zentrum München – German Research Center for Environmental Health, Munich, Germany. [8]Institute of Functional Epigenetics, Helmholtz Zentrum München – German Research Center for Environmental Health, Neuherberg, Germany. [9]Institute of Computational Biology, Helmholtz Zentrum München – German Research Center for Environmental Health, Neuherberg, Germany. [10]Berlin Institute of Health (BIH), MDC-Berlin, Berlin, Germany. [11]These authors contributed equally: Robert A. Beagrie, Christoph J. Thieme, Carlo Annunziatella. [12]These authors jointly supervised this work: Lonnie R. Welch, Mario Nicodemi, Ana Pombo. ✉e-mail: welch@ohio.edu; mario.nicodemi@na.infn.it; ana.pombo@mdc-berlin.de

achieves strong enrichment for contacts between regions containing active genes, enhancers and super-enhancers while requiring only a few hundred cells[3]. GAM has also been recently used for haplotype reconstruction and phasing of genetic variants, an essential prerequisite for detection of allele-specific analysis of chromatin contacts in non-model organisms or individuals with unknown haplotypes[4].

The original GAM protocol involves DNA sequencing of many individual thin nuclear slices (which we call nuclear profiles), each isolated in a random orientation from a different cell in the population. The principle behind GAM is that DNA loci that are physically close to each other in the nuclear space are present in the same nuclear profile more frequently than loci that are remote from one another. In the prototype version of GAM, a collection of thin (200 nm) cryosections were cut through a sample of mouse embryonic stem cells, before microdissection of single nuclear profiles into separate polymerase chain reaction (PCR) tubes, followed by lengthy manual preparation of sequencing libraries to determine the DNA content of each tube.

We now introduce several significant improvements to GAM. First, to reduce the hands-on time required for sequencing hundreds or thousands of nuclear profiles, we developed multiplex-GAM. In this variant of GAM, multiple independent nuclear profiles can be added into a single tube and then sequenced together, cutting down on both labor and reagent costs. Second, we optimized the protocol for DNA extraction from nuclear profiles such that it is now compatible with liquid dispensing robots, further reducing time and reagent volumes required to generate a GAM dataset. Third, we extended the SLICE statistical model for analysis of GAM data to cover a wider range of experimental parameters, including the addition of multiple nuclear profiles per tube. We also use the SLICE statistical model to determine optimal experimental parameters to aid the design of GAM experiments in different cells and organisms. Fourth, we expanded our GAM dataset on mouse embryonic stem cells from 408 to 1,250 cells, which we use for comparison with Hi-C. Finally, we show that many contacts are equally detected by both methods, but also identify method-specific contacts, especially those that involve simultaneous associations between three or more genomic elements. We show that GAM is a versatile methodology for mapping chromatin contacts that has several advantages over Hi-C (Supplementary Table 1). We also provide a framework to design GAM experiments that considers the depth of chromatin contact information required and minimizes data collection effort.

## Results

### Multiplex-GAM reduces sequencing costs and hands-on time

We previously published a GAM dataset of 408 single nuclear profiles (408 × 1NP) from mouse embryonic stem cells, in which each nuclear profile was isolated from a different nucleus into a single PCR tube (Fig. 1a, original-GAM)[3]. We showed that the number of times that particular genomic loci are found together in the same nuclear profile (their co-segregation) is a measure of their physical proximity in the original population of cells, with high co-segregation values indicating that the regions are close in space. Each nuclear profile contains only ~5% of the genome, and loci on different chromosomes are found together in less than 1% of nuclear profiles. We therefore reasoned that combining more than one nuclear profile into a single sequencing library would not reduce our ability to distinguish interacting from non-interacting loci (Fig. 1a, multiplex-GAM).

To test this idea, we used a dataset of 481 single nuclear profiles sequenced individually (481 × 1NP), which consists of 408 previously published samples[3] plus 73 additional single nuclear profile (1NP) datasets (Supplementary Table 2). To simulate multiplex sequencing of two or three nuclear profiles (2NP or 3NP), we combined 480 of the single nuclear profile datasets and generated 240 or 160 in silico GAM samples containing two or three nuclear profiles, respectively. We then re-calculated co-segregation matrices from these simulated

multiplex-GAM datasets and found that these were visually highly similar (Fig. 1b) and highly correlated (Extended Data Fig. 1).

To formally understand the effect of including several nuclear profiles in multiplex-GAM experimental designs and to optimize our experimental parameters, we extended SLICE, the statistical tool previously developed to infer non-random DNA interaction probabilities from locus co-segregation in GAM data (Extended Data Fig. 2)[3]. SLICE now considers the effects of number of nuclear profiles per GAM sample, nuclear ellipticity and nuclear profile thickness (Fig. 1c, Extended Data Fig. 3a–d and Supplementary Table 3). To determine the optimal parameters for collection of multiplex-GAM datasets in mouse embryonic stem cells, we applied the updated SLICE model to estimate the minimal number of tubes ($m^*$) required to detect chromatin contacts in different experimental designs (for example, different numbers of nuclear profiles per GAM sample; Supplementary Note). In general, multiplex-GAM performs similarly to original-GAM, but can require an increased number of nuclear profiles to detect the weakest contacts (including inter-chromosomal contacts), or to work at the highest genomic resolutions (that is, smaller window sizes; Fig. 1d).

Using the updated SLICE model, we calculate optimal experimental parameters for the application of GAM to a range of different organisms and cell types. Despite differences in ploidy and nuclear geometry between the selected cell types, we find that the minimum number of tubes ($m^*$) required to reach a given statistical power is approximately constant (requiring only ~200 tubes to detect contacts with a probability of interaction ($Pi$) ≥ 30%) provided that each sample is collected with the optimal number of multiplexed nuclear profiles (Extended Data Fig. 3e). Finally, we determined the optimal experimental parameters for producing a new multiplex-GAM dataset in mouse embryonic stem cells, and found that 3–10 nuclear profiles per GAM library is optimal. For example, a GAM dataset of ~250 libraries each multiplexed with three nuclear profiles (that is, obtained from a total of only 750 mouse embryonic stem cells) would be sufficient to sample contacts with interaction probabilities above 50% at a resolution of 30 kb across genomic distances >100 kb, while reducing reagent costs and experiment time by two-thirds (Extended Data Fig. 3f).

Next, we implemented several improvements to the original experimental pipeline for GAM data collection, including staining of cell profiles for better identification (Extended Data Fig. 4a). First, we screened for chemical stains compatible with both the direct visualization of the nucleus prior to microdissection and high-quality DNA extraction. We found that most DNA stains prevent subsequent extraction and/or amplification of DNA from nuclear profiles, probably because they bind too strongly or damage DNA (Extended Data Fig. 4b). We identified a cresyl violet stain that does not distinguish the cytoplasm from the nucleus, but greatly improves identification of cellular profiles during microdissection without affecting DNA extraction (Extended Data Fig. 4c,d). To estimate the frequency of cellular profiles that intersect the nucleus in a cresyl violet collection, we counterstained mouse embryonic stem cell cryosections with SYTO RNASelect and 4,6-diamidino-2-phenylindole (DAPI), and found that approximately three in four cellular profiles intersect the nucleus (Extended Data Fig. 4e,f).

To directly test the multiplex-GAM approach with our revised experimental pipeline, we collected a new batch of 249 multiplex-GAM sequencing libraries, each containing three nuclear profiles on average, from an independent biological replicate of mouse embryonic stem cells (Fig. 1e). The genomic coverage was comparable across different collection batches (18% of 40 kb windows are detected per nuclear profile on average; Extended Data Fig. 5a,b) and was consistent with the expected presence of three nuclear profiles per library on average (7% for 1NP data, 20% for 3NP in silico data). Comparison of normalized linkage matrices between the 249 × 3NP multiplex-GAM dataset and the 481 × 1NP original-GAM dataset indicated that local contact information is well preserved in multiplex-GAM (Fig. 1e).

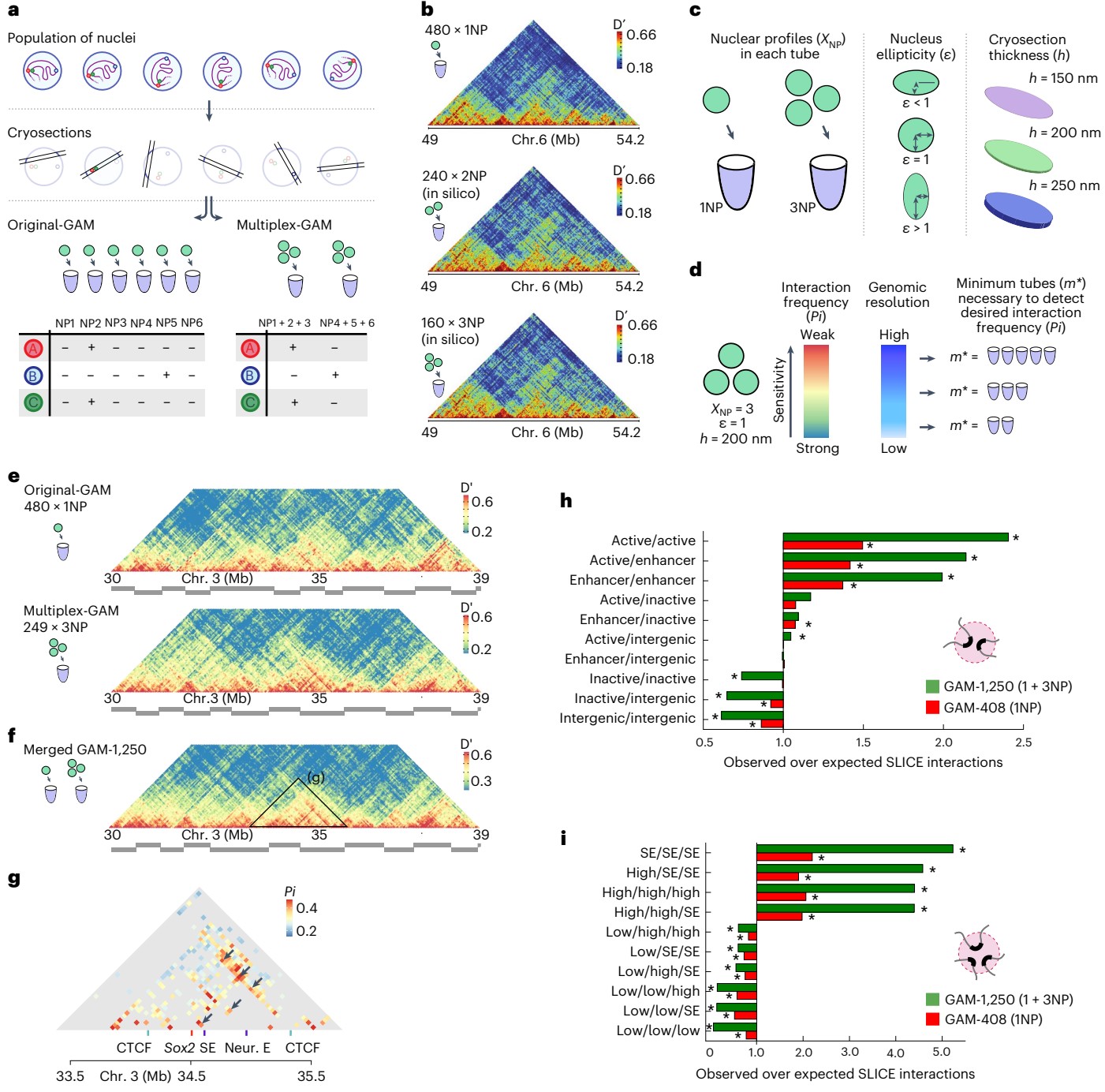

**Fig. 1 | An updated mouse embryonic stem cell chromatin contact map produced with multiplex-GAM. a**, In a standard GAM experiment, thin slices from individual nuclei (nuclear profiles, NPs) are isolated by cryosectioning and laser microdissection, before the DNA content of each slice is determined by next-generation sequencing. In a multiplex-GAM experiment, DNA from multiple NPs is extracted and sequenced together, reducing sequencing costs. **b**, Multiplex-GAM data constructed in silico by combining 1NP datasets at 40 kb resolution (chromosome (chr.) 6, 49–54 Mb). Contact maps were produced from single-NP data (top), in silico 2NP data (middle) and in silico 3NP data (bottom). D′, normalized linkage disequilibrium. **c**, The updated SLICE model accounts for the number of NPs multiplexed in each tube ($X_{NP}$), the nuclear ellipticity ($\varepsilon$) and the thickness of each NP ($h$). **d**, SLICE models can be used to guide the experimental design, for example to estimate the minimum number of tubes ($m^*$)

needed to achieve a given statistical power. **e,f**, Visualization of contacts centered at the *Sox2* locus (chr. 3, 30–39 Mb) in GAM 1NP data and 3NP data (**e**), and the combined GAM-1,250 dataset (**f**), all at 40 kb resolution. **g**, Significant pairwise interactions at 40 kb resolution identified by SLICE between functional elements in the *Sox2* locus, including the *Sox2* gene and its closest super-enhancer (SE). The arrows indicate previously identified interactions between these elements[38]. Neur. E, neuronal enhancer. **h,i**, Enrichment analysis of pairwise interactions (**h**) and triplet interactions (**i**) identified by SLICE involving active, inactive, intergenic or enhancer regions (**h**) and topologically associating domains that are highly transcribed (high), lowly transcribed (low) or that overlap super-enhancers (**i**). for the GAM-1,250 dataset and the original GAM-408 dataset. Statistically significant enrichments or depletions (those falling outside 95% of randomized observations after Bonferroni correction) are marked by an asterisk.

The 249 × 3NP dataset had a detection efficiency (probability of detecting any given genomic window) of 89% at 40 kb, and 80% of windows were detected at least 40 times. We therefore concluded that the quality of the 249 × 3NP dataset was at least as good as the 481 × 1NP dataset, which had a detection efficiency of 93% at 40 kb resolution and 80% of windows were detected at least 28 times.

We next considered the possibility of merging the 1NP and 3NP datasets. We first confirmed in silico that combining 1NP and 3NP libraries does not reduce the quality of the dataset (Extended Data Fig. 5c). We therefore merged the experimental 1NP and 3NP datasets to create a combined GAM dataset spanning a total of 1,250 nuclear profiles, each from a different cell (Fig. 1f). To confirm the increased statistical power of the combined 1 + 3NP dataset, we used SLICE to identify interacting regions and compared them with those obtained with the original 408 × 1NP dataset[3] (Fig. 1g). We detect a greater number of interacting regions using the deeper 1 + 3NP dataset compared with the published 1NP data for both pairwise (Fig. 1h and Extended Data Fig. 5d) and three-way interactions (Fig. 1i), further confirming that the most prominent interactions found in mouse embryonic stem cells involve active and enhancer genomic regions. The 1 + 3NP dataset also enabled detection of 4,711 significant interactions with a 10% false discovery rate threshold (Extended Data Fig. 5e and Supplementary Table 4).

One of the key aims of genome-wide 3D chromatin folding assays is the detection of topologically associating domains (TADs)[5]. We compared TAD boundary calls between GAM, bulk Hi-C[6] and single-cell Hi-C[7], and found that the three approaches detect a similar set of boundaries (Extended Data Fig. 6a–d). Boundaries detected by all methods tend to have stronger insulation than boundaries detected by only one method. Others have reported a similar overlap of TADs called from the same dataset by different algorithms[8]; thus, these unique TADs are likely to reflect inherent method-dependent variability. The distribution of previously described features enriched at TAD boundaries was similar for boundaries common to Hi-C and GAM, although a few epigenetic features were not found enriched in the small number of boundaries unique to GAM (123; Extended Data Fig. 6e).

## Identification of differential and common contacts

GAM detects far more contacts at larger genomic distances than Hi-C, such as megabase-range contacts between super-enhancers, validated by single-cell fluorescence in situ hybridization (FISH) experiments[3]. In silico modeling of Hi-C and GAM data has shown that GAM performs better than Hi-C at capturing real distances (Spearman correlation: −0.89 for Hi-C and −0.99 for GAM)[9]. To investigate genome-wide differences between GAM and Hi-C in an unbiased fashion, we developed a method for directly comparing matrices derived from the two methods. For these analyses, we considered contacts between loci separated by ≤4 Mb, given that the fidelity of Hi-C decreases at larger genomic distances. The selected genomic length scale is useful in most current applications of chromatin contact mapping; in particular, it is sufficient for the detection of enhancer–promoter contacts in most instances.

Given that GAM and Hi-C data have very different numerical distributions, we first applied a distance-based z-score transformation to both datasets to address the distance decay (Fig. 2a, rows 1 and 2). We then subtracted the two normalized matrices (row 3) and extracted the most divergent contacts, that is, those for which the difference between the two matrices was greater than the 5% extremes defined by a fitted normal distribution (row 4). We refer to these most differential contacts as GAM-specific or Hi-C-specific contacts. To explore the contacts that are well detected by both GAM and Hi-C, we also established a set of strong-and-common contacts by selecting the 10% strongest contacts from among the least differential contacts (with z-score delta <1.0; row 5).

We verified that the GAM-specific and Hi-C-specific contacts have similar distance decays (Extended Data Fig. 7a), and most are also found with alternative normalization methods (Extended Data Fig. 7b). We

also verified that the GAM-specific contacts selected have high intensity in GAM and low intensity in Hi-C, and vice versa for Hi-C-specific contacts (Extended Data Fig. 7c). Furthermore, we determined whether the most prominent contacts captured with SLICE from GAM data, or with Fit-Hi-C from Hi-C data, were differentially detected between the two methods (Fig. 2b and Extended Data Fig. 7c). Whereas the strongest GAM contacts detect a proportion of Fit-Hi-C contacts, the strongest Hi-C contacts are strongly depleted from the most prominent SLICE contacts detected in GAM data (Fig. 2c). Finally, we investigated the detectability of the genomic windows involved in Hi-C- or GAM-specific contacts, and found that GAM-specific contacts tend to originate from windows with the strongest detectability whereas Hi-C-specific contacts tend to involve fewer ligation events (Extended Data Fig. 7d). Strong-and-common contacts are often found in the 20% strongest Hi-C and/or GAM contacts (Extended Data Fig. 7c), and have a distance decay that peaks at 300–1,000 kb (Extended Data Fig. 7e). Many of the strong-and-common contacts are also detected by SLICE analyses of GAM data and/or by Fit-Hi-C analysis of Hi-C data (Fig. 2c).

## Multiplex-GAM detects many active contacts missed by Hi-C

To assess whether the contacts differentially detected by GAM or Hi-C have important biological roles, we investigated whether they were enriched for particular genomic features (Fig. 3a). We created a dataset of features including repeat elements, heterochromatin marks, transcription factor binding sites, RNA polymerase II and transcription-related histone marks (Supplementary Tables 5 and 6). We then counted the number of contacts in each category (GAM-specific, Hi-C-specific, strong-and-common) between each possible pair of features (for example, CTCF–CTCF, p300–Nanog, and so on), and looked for feature pairs overrepresented (enriched) or underrepresented (depleted) from GAM-specific or Hi-C-specific contacts relative to distance-matched random backgrounds (Extended Data Fig. 8a and Supplementary Table 7).

We found most feature pairs more frequently in the sets of specific contacts than in the genomic background (Fig. 3b and Supplementary Tables 8 and 9). Most of the feature pairs show a stronger enrichment in GAM-specific contacts than in Hi-C-specific contacts, whereas only a small subset of feature pairs are more frequent in Hi-C-specific contacts. To prioritize the most important of feature pairs that best discriminate GAM- and Hi-C-specific contacts, we used a random forest method (Extended Data Fig. 8b,c). Of the 10 feature pairs with the highest discriminatory power, six involve the known architectural factor CTCF, interacting with active features (RNA polymerase, p300, enhancers or Oct4). Interestingly, CTCF–CTCF and CTCF–heterochromatin contacts were also enriched in GAM-specific contacts. By contrast, heterochromatin regions (that is, those marked by H3K9me3 or H4K20me3) were the only features most enriched in the set of Hi-C-specific contacts (Fig. 3c).

As an example, we observe an extensive network of GAM-specific contacts at the 5′ side of the 11qC locus, spread throughout a gene-dense region that includes multiple genes with suggested roles in gene regulation (*Ints2, Med13, Supt4h1, Coil*) and mouse embryonic stem cell pluripotency (*Vezf1, Msi2, Trim25, Nog*; Fig. 3d). By contrast, the 3′ side of the 11qC locus harbors a gene-poor region involved in a large number of Hi-C-specific contacts. Given that the 40 kb windows forming contacts overlap with multiple different genomic features, we measured the co-occurrence of feature pairs using UpSet plots (Fig. 3e). Five of the 10 most frequent groups of feature pairs identified from GAM-specific contacts overlap at least six different feature pairs linking CTCF and/or active chromatin, while only one such group appears in the top 10 for Hi-C-specific contacts. These results suggest that GAM-specific contacts are strongly enriched for a specific subset of CTCF–CTCF contacts that co-occur with enhancers and active genes and which are underestimated in Hi-C data. In contrast, CTCF–CTCF contacts that overlap no other annotated features are the third most

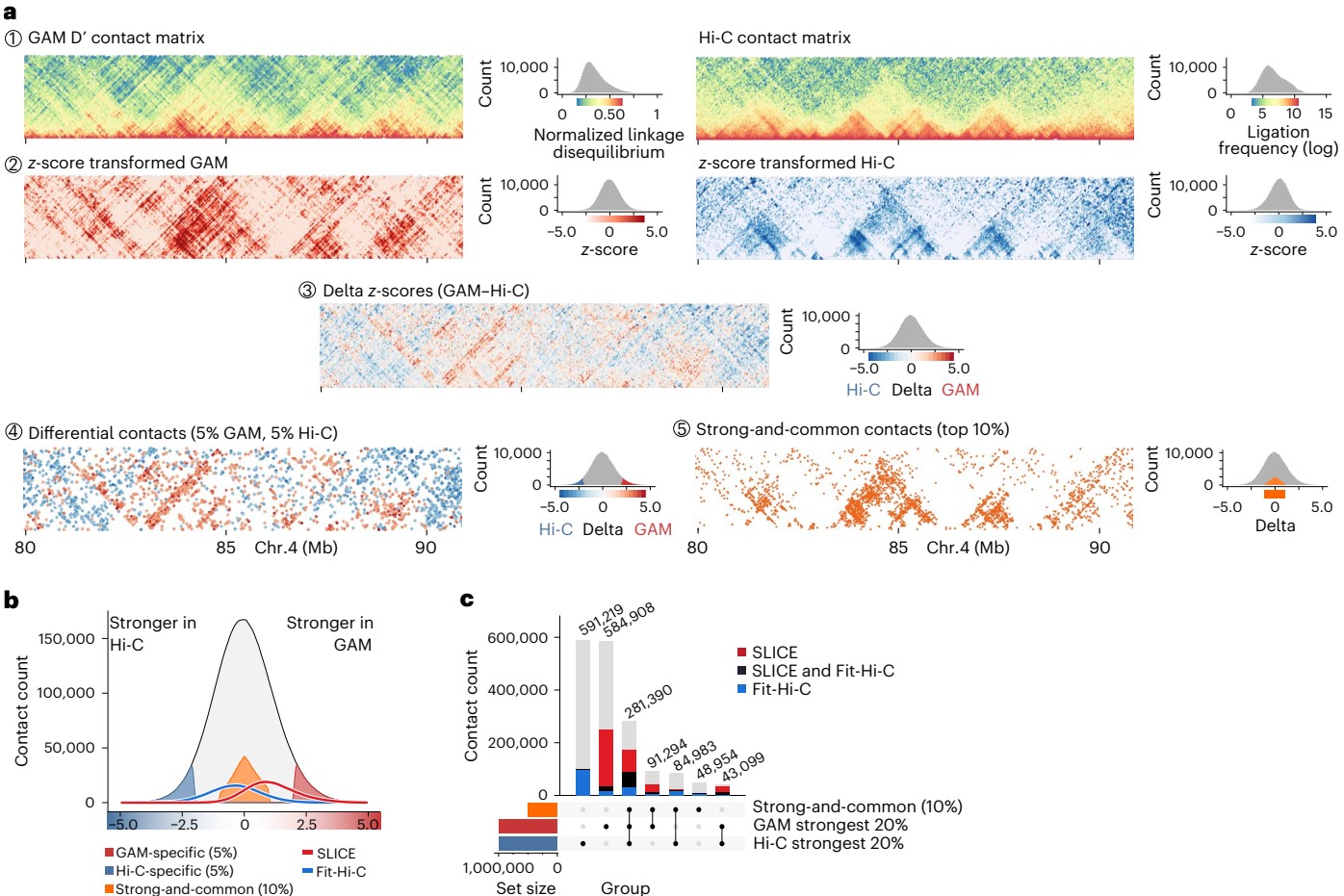

**Fig. 2 | GAM and Hi-C show strong, method-specific contacts. a**, Strategy to assess differences and similarities between GAM and Hi-C contact maps (chr. 4, 80–90 Mb). GAM and Hi-C contact data have different distributions (1); therefore, contacts at the same genomic distance undergo $z$-score transformation (2). Hi-C $z$-scores are then subtracted from GAM $z$-scores to generate a delta $z$-score matrix (3), from which we extract the 5% most differential contacts between GAM and Hi-C (GAM-specific or Hi-C-specific; 4) and the top 10% of contacts common for both methods (strong-and-common; 5). **b**, Interactions identified by SLICE or by Fit-Hi-C shown in the distribution of delta $z$-scores. **c**, Overlap of contacts co-occurring in combinations of the top 20% strongest contacts from GAM and Hi-C, and the 10% strong-and-common set. Intersection groups are colored by fraction supported by SLICE or Fit-Hi-C.

frequently detected set of contacts found in the strong-and-common contacts equally detected by both methods.

### Active regions are underrepresented in Hi-C data

Having identified striking enrichments for specific genomic features among GAM- and Hi-C-specific contacts, we investigated whether certain features might be generally poorly detected by either method. To identify such potential blind spots, we developed an approach that counts the number of GAM-specific, Hi-C-specific and strong-and-common contacts formed by each window and investigated whether specific genomic regions were typically more involved in GAM-specific or Hi-C-specific contacts or vice versa (Fig. 4a). Surprisingly, we find that blind spot windows are fairly abundant, as shown by the flares of method-specific contacts at specific genomic regions (Fig. 4b). Furthermore, blind spot windows are often clustered in specific regions of the linear genome.

To investigate the properties of the genomic regions underrepresented in GAM- or Hi-C-specific contacts, we selected the genomic windows in the top deciles of method-specific, or strong-and-common contacts. We found that the genomic windows that form many GAM-specific contacts (here called GAM-preferred regions) contain more genes and have higher transcriptional activity (Fig. 4c,d) than genomic regions that form many Hi-C-specific contacts (Hi-C preferred

regions), which in turn are more frequently associated with the nuclear lamina[10] (Fig. 4e). GAM-preferred regions also tend to be occupied by CTCF, p300, certain mouse embryonic stem cell transcription factors, RNA polymerase II (especially the elongating, S2p form), enhancers and super-enhancers, and are often classified as compartment A (Fig. 4f). By contrast, Hi-C-preferred regions showed a slight enrichment for the heterochromatin-associated histone marks H4K20me3 or H3K9me3, and are more frequently classified as compartment B. Tracks of all genomic features considered are also shown across an 80 Mb region in chromosome 8 in a genome browser visualization (Extended Data Fig. 9a), and their co-occurrence in the same genomic windows highlights the presence of CTCF, transcriptional activity features, including super-enhancers, in GAM-preferred regions.

### Complex contacts cause discrepancies between GAM and Hi-C

We considered whether the enrichment for active features (active genes, transcription factors, polymerase, enhancers and compartment A) in contacts preferentially detected by GAM could be due to different levels of contact complexity, that is, to interactions with many simultaneous interacting partners (Fig. 5a). Complex interactions have been predicted to be underestimated in Hi-C datasets because the ligation step allows only for the measurement of two interacting partners per restriction fragment in each cell where the contact is established[11].

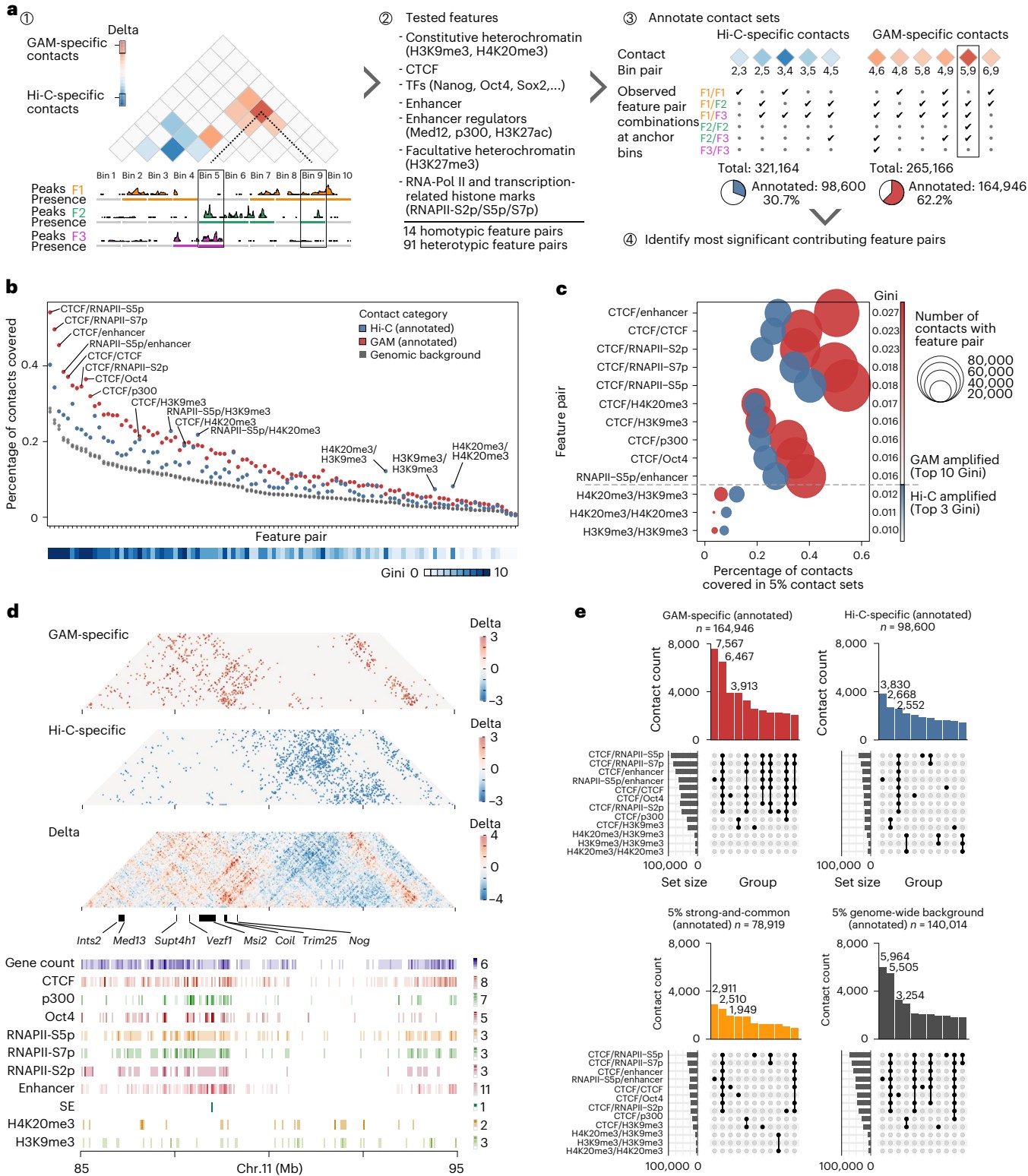

**Fig. 3 | Gene activity at anchor points distinguishes differential contacts.**
**a**, Schematic for detecting feature pair enrichments in GAM- or Hi-C-specific contacts (red and blue, respectively). Each contact is defined by two genomic anchor points that we categorized as either positive or negative for peaks of the respective feature (1). We assessed 105 feature pairs (2) and quantified feature occurrences at the anchor points, filtering out contacts with no feature pairs (3) to identify most distinctive feature combinations between GAM-specific and Hi-C-specific contacts (4). TF, transcription factor. **b**, Top: frequency of feature pairs in annotated GAM-specific and Hi-C-specific contacts ranked by their presence in the genome. Bottom: heatmap track for the Gini impurity score using the

random forest classification, which was trained to discriminate GAM-specific and Hi-C-specific contacts. **c**, Feature pairs with the highest discriminatory power. Top 10 by mean decrease of Gini impurity, all amplified in GAM (top) and the top 3 feature pairs with the strongest amplification (highest abundance in Hi-C relative to GAM; below dashed line) relative to the abundance of the feature pair in the contact sets. **d**, Top: contact matrices of GAM-specific and Hi-C-specific contacts at an example locus (chr. 11: 85–95 Mb). Bottom: locations of distinctive features. **e**, UpSet plots quantifying the co-occurrence of enriched feature pairs selected in **c** for contacts of each subgroup.

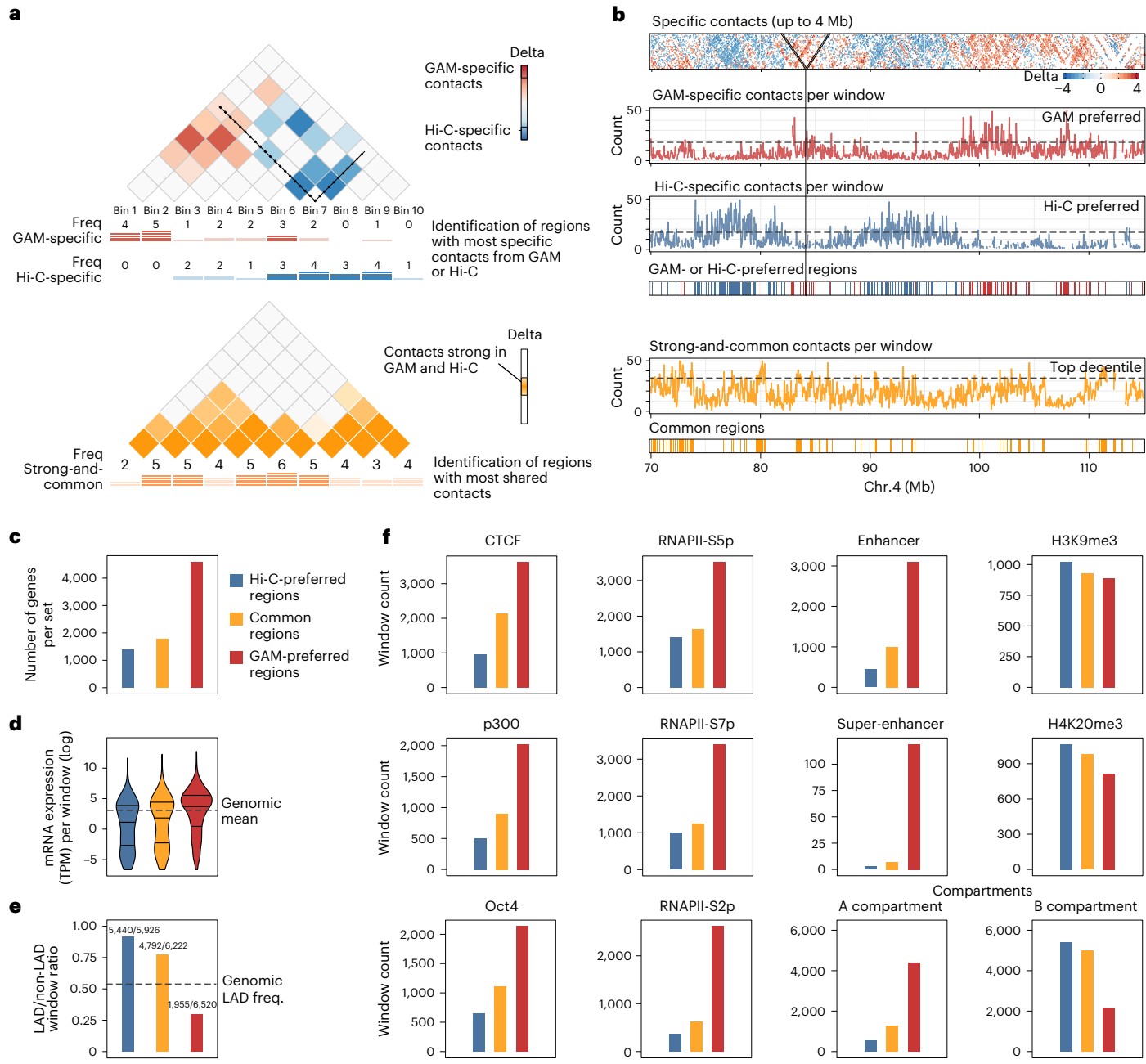

**Fig. 4 | Genomic windows enriched for Hi-C- and GAM-specific contacts are clustered in the genome. a**, Strategy for identifying genomic regions forming many contacts specific to either GAM or Hi-C. We counted how often a genomic region was an anchor point in the set of GAM-specific contacts, Hi-C-specific contacts or strong-and-common contacts. The 10% of genomic windows with the highest absolute difference between the number of GAM-specific and Hi-C-specific contacts were classified as Hi-C-preferred regions or GAM-preferred regions, respectively. Similarly, the top 10% of strong-and-common contacts were used to define common regions that participate to a similar extent in strong contacts seen in both methods. **b**, Example region on chromosome 4 showing the delta *z*-score matrix (top) and clusters of method-specific preferences designated as Hi-C-preferred regions, GAM-preferred regions or common regions (bottom). **c**–**f**, Characteristics of Hi-C-preferred, common and GAM-preferred windows. **c**, Total number of genes present in windows of each category. **d**, Transcriptional activity (log₂-scaled transcripts per million per gene, TPM) of all genes per category. Bars mark the 25%, 50% and 75% quantiles. **e**, Proportion of windows in lamina-associated domains (LADs) for each category. Numbers on top of the bars are the number of LAD windows and the total number of windows for each category. **f**, Numbers of genomic windows in each category that overlap a range of features or are classified as compartment A or B.

To investigate the relationship between interaction complexity and method-specific blind spots, we used SLICE to calculate the probability of interaction for all possible sets of three 1 Mb windows lying on the same chromosome (that is, the $Pi_{ABC}$ for all possible triplets of loci A, B and C)[3]. We find that windows in the A compartment indeed form more triplets than windows in the B compartment (Fig. 5b and Extended Data Fig. 10). Interestingly, GAM-preferred regions formed more triplets than common or Hi-C-preferred regions, even when comparing within the same compartment. Regions with active chromatin marks formed more triplets than regions marked by heterochromatin, with the strongest effect seen for the elongating, S2-phosphoisoform of RNA polymerase II and for super-enhancers (Fig. 5c), in line with our

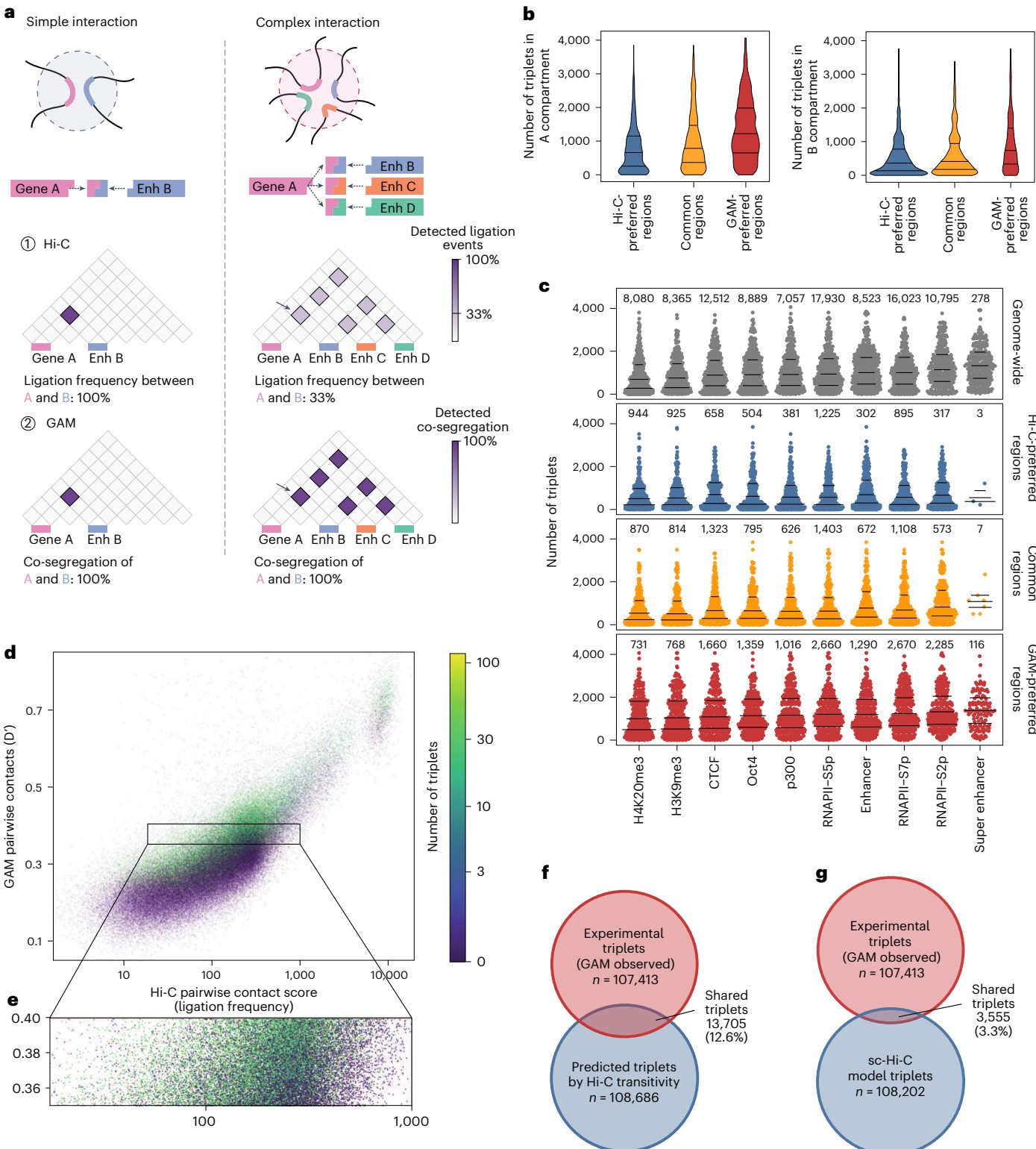

**Fig. 5 | Higher complexity contacts are more frequently captured by GAM.**
**a**, Interaction complexity: simple interactions involve only a few genomic regions, while complex interactions involve many genomic regions at once. In Hi-C, ligation events connect two pieces of DNA; therefore, pairwise contacts can be directly determined while measurement of higher-order contacts may be affected by competing ligation events resulting in dilution effects (1, arrow). Co-segregation determined through GAM is not affected by the number of elements (2). **b**, Number of triplets formed by Hi-C-preferred, common or GAM-preferred regions in the A compartment (left) or the B compartment (right). **c**, Number of triplets formed by genomic windows that are positive for a feature,

shown genome-wide and within Hi-C-preferred, common or GAM-preferred regions. Bars mark the 25%, 50% and 75% quantiles; labels give the total number of windows in each group. **d**, Relationship between pairwise Hi-C ligation frequency, pairwise GAM normalized linkage and the number of triplets. **e**, Hi-C ligation frequency of 1 Mb pairwise contacts that have a similar GAM normalized linkage, colored by number of triplets (zoomed subset of **d**). **f**, Overlap of triplet contacts identified by SLICE from GAM data and inferred from Hi-C by using the strongest 2% based on transitivity. **g**, Overlap of triplets found by SLICE in GAM data and triplets derived from 3D models from single-cell (sc)-Hi-C.

previous work that identified long-range chromatin contacts between super-enhancers and actively transcribed genomic regions across tens of megabases[3]. These results suggest the existence of abundant chromatin contacts in which many active regions interact simultaneously, which are commonly overlooked by ligation-based methods, but are readily detected by GAM and FISH[3].

Finally, we examined whether high interaction complexity artificially deflates pairwise contact probability as measured by Hi-C, given that each DNA fragment is predicted to pick up a given interacting partner with lower probability in complex contacts than in simple contacts (Fig. 5d)[11]. We correlated pairwise contacts from GAM and Hi-C at a resolution of 1 Mb and found that regions with equivalent strength of pairwise contacts in GAM had a broad range of ligation frequencies in Hi-C. Regions that form many triplets (that is, that are more complex) had lower contact strength in Hi-C data. Conversely, regions that form few triplets had higher contact strength in Hi-C (Fig. 5d,e), demonstrating that complexity explains some of the divergence in contact frequencies measured by Hi-C or GAM. Notably, this effect also undermines attempts to predict the formation of complex interactions from Hi-C based only on the transitivity of pairwise contacts. For example, if locus A interacts with B, B with C, and A with C, simple transitivity predicts the formation of an A-B-C triplet detected with a frequency at most as high as at the lowest pairwise contact frequency between any of the locus pairs. Direct comparisons between the triplets identified in GAM data (top 2% most statistically significant) with the top 2% Hi-C triplets, which are computed assuming transitivity, show little overlap, with less than 15% of true triplets detected by GAM coinciding with triplets predicted based on transitivity from pairwise Hi-C maps (Fig. 5f) or single-cell Hi-C maps[12] (Fig. 5g). Therefore, transitivity of pairwise contacts cannot be used to infer multiway contacts.

## Discussion

The three-dimensional structure of the nucleus is inextricably linked with its functional roles, including gene regulation, DNA replication and the DNA damage response. Consequently, molecular techniques for measuring the 3D folding of chromatin inside the nucleus have been instrumental in advancing our understanding of nuclear function over the past decade[2]. Here, we have developed multiplex-GAM, a new variant of genome architecture mapping that enables faster and more cost-effective analysis of chromatin folding genome-wide than the original version[3]. We also expand the mathematical model SLICE by incorporating new experimental parameters (number of nuclear profiles per sample, nuclear ellipticity and cryosection thickness). Finally, we use the larger GAM dataset containing information from 1,250 mouse embryonic stem cells for detailed comparisons of the contacts captured by GAM and Hi-C, the most frequently used genome-wide method for chromatin contact analysis[13].

We find that GAM and Hi-C detect similar TADs, large folded domains that are thought to constrain gene regulatory elements and form a fundamental unit of chromatin organization[6,14,15]. Many strong contacts, including a large proportion of CTCF-mediated loops, are also detected by both methods. By careful examination of finer-scale differences, we identify that chromatin contacts given more weight by GAM frequently connect genomic loci bound by enhancers, key mouse embryonic stem cell transcription factors, RNA polymerase II and CTCF, whereas contacts that feature more prominently in Hi-C matrices connect regions marked by the heterochromatin-associated histone modifications H3K9me3 and H4K20me3.

We looked for regions of the genome that consistently form more contacts in GAM datasets than in Hi-C datasets and found that these regions are located in large genomic regions bound by the same activating transcription factors identified in the GAM-specific contacts. In our previous work, super-enhancers were the genomic regions most enriched in complex, multi-partner interactions, together with the most actively transcribed regions[3]. We now extend this finding to

show that the contacts underestimated in Hi-C often involve regions that form more complex interactions in GAM. Theoretical work has previously suggested that ligation-based methods, such as Hi-C, underestimate contacts between multiple partners, given that ligation captures only two or a few contact partners at a time[11]. Our results here show that ligation frequencies measured by Hi-C are systematically lower between regions that form complex interactions, and provide experimental evidence to support the effect of ligation on the underestimation of complex contacts.

Ligation is not the only potential source of difference between the two methods, given that GAM and Hi-C also make use of quite different fixation protocols. The choice of fixation protocol has been shown to affect the proportion of informative ligation events between different chromatin conformation capture experiments[16], and it may also influence the contacts of genomic regions with different protein composition and/or compaction in a single experiment[17,18]. The digestion of nuclear chromatin necessary for preparing Hi-C libraries has also been shown to disrupt nuclear structure[19], whereas GAM uses fixation protocols specifically chosen to maximize the preservation of nuclear architecture and retention of nuclear proteins[20] and RNAs[21]. Ultimately, formaldehyde fixation remains a 'black box' and will continue to complicate interpretation of the most widely used methods for measuring chromatin structure (including microscopy methods such as FISH)[22]. Live-cell imaging methods circumvent the need for fixation and will provide valuable orthogonal data, but these methods currently require recruitment of large numbers of fluorophores, which may themselves influence folding[23]. Variants of chromatin conformation capture have also been reported with a different order of steps[24] or that do not use fixation, but omission of the fixation step entirely has a variable impact on signal-to-noise ratio[25,26]. Ultimately, it should eventually be possible to shed light on the effect of fixation by extending GAM to unfixed nuclei through sectioning of vitrified samples.

Another factor that may influence method-specific contacts is data processing. It has recently been shown that Hi-C detects fewer contacts between regions of condensed chromatin due to a lower accessibility of these regions to restriction enzyme digestion[27]. However, matrix-balancing algorithms commonly used to normalize Hi-C data can overcorrect for this effect, leading to an aberrantly high frequency of contacts between condensed domains. Consistent with these results, we find that regions of the genome that consistently form more contacts in normalized Hi-C are enriched for heterochromatin marks, and link two regions with low detectability (that is, those most likely to be overcorrected by matrix balancing). We have found the bias in raw GAM datasets to be uniformly lower than that found in raw Hi-C[3] and expect that improved normalization algorithms will bridge some of the current divergences between the two methods[27–30].

Our work underscores previous findings that complex, simultaneous interactions between many genomic regions are a pervasive and little-studied feature of mammalian genome architecture[3,31], although their overall prevalence is still a subject of debate[32]. Enhancer-binding transcription factors and RNA polymerase II have both been reported to form nuclear clusters that could serve as nucleating agents for such multi-partner interactions[33,34]. More recently, there has been a surge of interest in phase-separated nuclear bodies, which are suggested to facilitate high local concentrations of chromatin-interacting proteins and/or transcriptional regulators[35]. The clear expectation is that these condensates should bring together multiple interacting genomic partners, in much the same way as ribosomal DNA repeats are brought together in the nucleolus[36]. Heterochromatin has also been reported to form phase-separated condensates[37]; however, we find these regions to have lower-complexity specific interactions, potentially highlighting a shorter-range role for these interactions.

In conclusion, our development of multiplex-GAM, an improved protocol for rapid, cost-effective generation of GAM datasets, enabled us to obtain a deeper GAM dataset for mouse embryonic stem cells and to explore the similarities and differences between GAM and Hi-C.

Reassuringly, the two methods paint broadly similar pictures of nuclear architecture, in particular the distribution of TADs, the segregation of nuclear chromatin into A and B compartments and the importance of CTCF for shaping chromatin interactions. There are differences, however, with GAM detecting more, stronger and more complex contacts between active chromatin regions, and across longer distances, and Hi-C emphasizing less-complex contacts within silent chromatin. These results highlight the utility of GAM for studying contacts of potential gene regulatory functions, particularly in human disease, where such contacts may be formed only in rare cell populations inaccessible to population Hi-C. We have recently applied multiplex-GAM to different neuronal subtypes in brain tissues, and discovered unforeseen events of extensive chromatin decondensation at long neuronal genes, and abundant cell-type specific contacts that contain differentially expressed genes and accessible regulatory elements spanning several megabases[30]. GAM requires only a few hundred cells, which is of particular relevance to human genetics, where researchers need to assay the 3D contacts made by disease-linked sequence variants in specific, often rare cell types impacted by the disease (for example, neuronal subtypes in neurodegenerative diseases).

## Online content

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

## Methods

### Identification of cellular profiles not intersecting the nucleus

Cryosections were incubated in 2.5 µM SYTO RNASelect solution in PBS (ThermoFisher [S32703]) for 20 min at room temperature (hereafter 17–22 °C), followed by a 5 min wash in PBS. After incubation, the cells were counterstained with 0.5 µg ml⁻¹ DAPI in PBS, and then rinsed in PBS and water. Coverslips were mounted in Mowiol 4–88 solution (Merck 81381) in 5% glycerol, 0.1 M Tris-HCl, pH 8.5.

### Cryosection staining for GAM

**Eosine.** Cryosections were washed (three times, 15 min in total) in PBS, rinsed in water and incubated for 3 min in a 1% Eosine Y solution (Merck 230251; dissolved in 1% glacial acetic acid in water) or 0.5% Eosine Y solution (dissolved in 0.25% glacial acetic acid, 70% ethanol in water). After incubation the cryosections were briefly washed three times in water and air dried for 5 min at room temperature.

**Propidium iodide.** Cryosections were washed (three times, 15 min in total) in PBS, rinsed in water and incubated for 10 min in a 10 µg ml⁻¹ propidium iodide solution (Sigma-Aldrich P4864 diluted in PBS). After incubation the cryosections were briefly washed three times in water and air dried for 5 min at room temperature.

**Crystal violet.** Cryosections were washed (three times, 15 min in total) in PBS, rinsed in water and incubated for 10 min in a 1% crystal violet water solution (Merck V5265). After incubation the cryosections were briefly washed three times in water and air dried for 5 min at room temperature.

**Cresyl violet.** Cryosections were washed (three times, 15 min in total) in PBS, rinsed in water and incubated for 6 min in a 0.1% cresyl violet (Sigma-Aldrich, C5042) water solution. After incubation the cryosections were briefly washed three times in water and air dried for 5 min at room temperature.

**SYBR Gold.** Cryosections were washed (three times, 15 min in total) in PBS, rinsed in water and incubated for 10 min with 1:1,000 or 1:5,000 dilution of SYBR Gold (ThermoFisher [S11494]) in water. After incubation the cryosections were briefly washed three times in water and air dried for 5 min at room temperature.

### Cell lines

Sox1-green fluorescent protein (Sox1-GFP) knock-in (cell line 46C) mouse embryonic stem cells derived from the parental E14tg2a line were used in this study[39]. Identity was confirmed at the time of cryoblock creation by morphology and by confirming GFP expression after neural differentiation. Cells were routinely tested for *Mycoplasma* contamination.

### Updated GAM protocol

Mouse embryonic stem cells were grown and cryoblocks prepared as previously described[3]. Cryosections of 220 nm (green) were cut with glass knives using a Leica FC7 ultracut cryotome, collected in sucrose droplets (2.1 M in PBS) and transferred to steel frame PEN (polyethylene naphthalate) membrane slides (Leica) for ultraviolet treatment for 45 min prior to use. Slides were washed in sterile-filtered (0.2 µm syringe filter) 1× PBS (three times, 5 min each), then with sterile-filtered water (three times, 5 min each). Cresyl violet staining was performed with sterile-filtered cresyl violet (1 % w/v in water, Sigma-Aldrich, C5042) for 10 min, followed by two washes with water (30 s each) and air dried for 15 min. Nuclear profiles were laser microdissected into adhesive 8-strip laser capture microdissection collection caps (Zeiss AdhesiveStrip 8C opaque 415190-9161-000), with four profiles dissected into each cap. Caps were stored at −20 °C until whole genome amplification.

Whole genome amplification of DNA from microdissected nuclear profiles was performed with the Sigma WGA4 kit using a liquid handling robot (Microlab STARlet, Hamilton). We note that several consecutive Sigma WGA4 kits stopped working in 2017 for GAM data production, and we currently recommend a more affordable in-house whole genome amplification protocol[30]. A total of 14.5 µl lysis and fragmentation master mix (13 µl H₂O, 1.4 µl lysis and fragmentation buffer, 0.09 µl proteinase K) was added to each well of a 96-well plate, caps with microdissected material were used to close the wells and then the plate was inverted and centrifuged upside down at 3,000 ×g for 2 min such that the fragmentation master mix was collected in the cap. Plates were incubated upside down for 4 h at 50 °C then inverted and centrifuged the right way up at 3,000 ×g for 2 min to collect the extracted DNA in the bottom of the well. Samples were then heat inactivated at 99 °C for 4 min then cooled on ice for 2 min. A total of 4.95 µl library preparation master mix (3.3 µl library preparation buffer, 1.65 µl library stabilization solution) was added to each sample, incubated at 95 °C for 2 min and cooled on ice for 2 min then centrifuged at 3,000 ×g for 2 min. A total of 4.5 µl library preparation enzyme (diluted threefold with H₂O) was added to each tube; then samples were incubated at 16 °C for 20 min, then 24 °C for 20 min, 37 °C for 20 min and 75 °C for 5 min. Finally, 85 µl amplification master mix (11 µl amplification buffer, 66.5 µl H₂O, 7.5 µl whole genome amplification polymerase) was added to each tube, and the samples were amplified by PCR (initial denaturation at 95 °C for 3 min, then 24 cycles of denaturation at 95 °C for 30 s and annealing–extension at 65 °C for 5 min).

Amplified DNA was purified using Ampure XP beads (Beckman Coulter, [A63880]). The beads (61.5 µl) were mixed with 77 µl amplified sample in a fresh 96-well plate and incubated at room temperature for 5 min. The plate was placed on a magnetic stand for 5 min; then the supernatant was discarded and the beads were washed twice with 200 µl freshly prepared 80% ethanol. After the second ethanol wash was discarded, the beads were air dried for 5 min and then resuspended in 45 µl H₂O and incubated at room temperature for 5 min. The plate was then placed on a magnetic stand and the supernatant transferred to a fresh 96-well plate, ready for next-generation sequencing library preparation.

Libraries were prepared using the Illumina Nextera library preparation kit following the manufacturer's instructions. The DNA concentration of the final libraries was determined using a Picogreen fluorescence assay (ThermoFisher), and libraries were pooled at equimolar concentration, ready for sequencing on an Illumina NextSeq machine.

### GAM data processing

Multiplex-GAM sequencing reads were aligned to the mouse mm9 genome assembly using Bowtie2 v2.1.0, and PCR duplicates were filtered using Samtools v0.9.0. Positive 40 kb windows were called by GAMtools v1.1.0 using a fixed read threshold of 4. The value of 40 kb was chosen for further analysis because it was the highest resolution at which the efficiency of detection (as calculated by SLICE) was greater than 80%, and >80% of 40 kb windows were detected at least 25 times in the multiplex-GAM dataset. Normalized linkage disequilibrium (D′) matrices at 40 kb genomic resolution were generated by GAMtools[40]. Further data analysis was carried out using Python v.3.7.

### SLICE analysis

To convert pair or triplet co-segregation frequencies to interaction probabilities ($Pi$), we computed the segregation probabilities $v_i$ for a single locus under an assumption of spherical shape, with an average nuclear radius $R$ (which was estimated using cryosection images as being equal to 4.5 µm)[3]. The co-segregation probabilities $u_i$ for pairs of loci in a not-interacting state have been estimated from GAM segregation data; for interacting loci we estimated co-segregation $t_i$ probabilities by assuming their physical distance as being less than

the slice thickness ($h \simeq 220$ nm). From linear combinations of these probabilities, using the 'mean field' approximation, we computed the probability of locus segregation in a nuclear profile for pairs ($N_{i,j}$) and triplets ($N_{i,j,k}$; Supplementary Note).

The expected number of nuclear profiles $M_i$ with 0, 1 or 2 loci is therefore computed from $N_{i,j}$ probabilities. From these, in turn, it is also possible to estimate the co-segregation ratio $M_1/(M_1 + M_2)$, that is, the fraction of non-empty tubes that have two loci. Given that the equations describing the tube content depend on the interaction probability $Pi$, the latter can be estimated by fitting the experimental value of co-segregation ratio (Supplementary Note). The same procedure has been used to estimate the probability of triplet interactions. Significant SLICE contacts are those with a co-segregation ratio greater than the 95th percentile of the expected distribution of co-segregation ratios for two non-interacting loci at the given genomic distance.

To apply SLICE to the merged multiplex-GAM dataset, we used a mean field approximation. It consists of introducing a non-integer number of nuclear profiles per tube, obtained as the average of the different numbers of nuclear profiles in the different datasets, weighted with the corresponding number of tubes (Supplementary Note).

### Creation of in silico merged multiplex-GAM data
Segregation tables (in which each row corresponds to a genomic window, each column to a GAM library, and the entries indicate the presence or absence of each window in each nuclear profile) were generated from 1NP GAM libraries. A new segregation table was then generated by randomly selecting two, three or four columns from the original table (that is 2/3/4× 1NP libraries), combining them into a single column such that the new column is positive if any of the original columns were positive, and removing the columns from the original table. This procedure was performed iteratively until all columns from the original table had been combined. The new, in silico combined table was then used for the calculation of normalized linkage disequilibrium matrices.

### SLICE enrichment tests
Enrichment of active/enhancer/inactive/intergenic windows in pairwise SLICE interactions and analysis of triplet SLICE interactions was carried out as previously described[3].

### SLICE false discovery rate thresholding
To identify the highest-confidence individual interactions, we used the R implementation of the Benjamini–Hochberg procedure to adjust the $P$ values for two-way interactions obtained from SLICE with a threshold of 0.1 (ref. [41]).

### TAD calling
We applied the insulation square method[42] to GAM matrices of normalized linkage disequilibrium scores and to Hi-C matrices of normalized ligation frequencies (GSE35156)[6] to exclude potential effects of using different TAD callers for GAM and Hi-C. We adjusted the insulation square method to also consider negative values from GAM normalized linkage disequilibrium and applied it to contact matrices at a resolution of 40 kb for each chromosome (using the parameters im mean, ids 50000, nt 0.1, insulationDeltaSpan 200000, yb 1.5, bmoe 3). Although the TAD sizes were not associated with the size of the insulation square for Hi-C data (reaching a plateau at a square size of around 500 kb), increased sizes of the insulation square produced larger TADs for GAM data. Here, we selected a window size of 400 kb for GAM and Hi-C data, which maximizes the agreement between the TAD sets and also to the hidden Markov model (HMM) TAD boundaries published for the Hi-C dataset[6]. Next, we used the merge command from bedtools v2.27.1 (ref. [43]) to check whether the obtained TAD boundaries were touching or overlapping, and merged the border ranges while retaining their maximum boundary score.

We obtained published single-cell Hi-C data for diploid mouse embryonic stem cells kept in serum media (GSE94489)[7] and created pseudobulk contact matrices at a resolution of 40 kb by pooling increasing subsets of 50 cells (50, 100, 150) and all 588 cells. Insulation profiles and TAD boundaries were computed using the insulation square method as described.

To check for overlapping boundary positions between two datasets we applied bedtools closest in both directions and considered boundaries as matched when their reported ranges were overlapping or touching (distance ≤1).

We checked for abundance of features at the TAD boundaries, centered at the boundary midpoints. For a given genomic mark, we analyzed the mean signal within 500 kb around the identified boundary midpoint in windows of 10 kb resolution using bedtools. We estimated the background by randomizing the boundary positions using chromosome-wise circular shifts.

### Generating peak and feature data
We mapped genomic and epigenomic read data to the NCBI Build 37/ mm9 reference genome using Bowtie2 v2.1.0 (ref. [44]). We excluded replicated reads (that is, identical reads mapped to the same genomic location) that were found more often than the 95th percentile of the frequency distribution of each dataset. We obtained peaks using BCP v1.1 (ref. [45]) in transcription factor mode or histone modification mode with default settings. A full list of all features analyzed in this study is given in Supplementary Tables 5 and 6. We computed mean counts of features for all genomic 40 kb windows using the bedtools window and intersect functions.

### PCA compartments
We computed eigenvalues and inferred compartments on GAM and Hi-C data as described[3,13] or used published compartment definitions[6].

### Identification of differential contacts
GAM and Hi-C use two different approaches to assess chromatin structure and measure underlying contact frequencies, which results in different distributions for GAM D′ values (continuous values resembling locus proximity in space) and log-scaled Hi-C frequency values (discrete cross-ligation counts)[9].

To compare contact intensities between the methods over the whole genome, and define strong contacts seen in both or at significantly different levels by either of the two methods, we developed a new method for identifying differential regions between the two matrices. To avoid amplification of spurious contacts due to potential undersampling and zero inflation, we limited our analysis to a 4 Mb genomic distance. From GAM contact data at 40 kb resolution, we removed all contacts with negative D′ values. We also excluded all contacts established between potentially oversampled or undersampled genomic windows. Here, we used the percentage of slices with a positive window (window detection frequency) as a proxy for detectability and removed windows with a window detection frequency of less than 5% or above 10%. For the Hi-C data, we excluded all contacts for which zero ligation events were detected. All contacts excluded from either dataset were not considered in the definition of differential contacts.

To compare the contact intensities from GAM and Hi-C, we evaluated a number of linear transformations, namely $z$-score transformation, observed over expected scores and rank transformation. For every chromosome, we applied $z$-scores and observed over expected transformation to GAM and Hi-C contacts at a given distance $d$. We found that the resulting intensity distributions of the delta matrices can be parameterized with very good fit to a normal distribution for $z$-scores and a logistic distribution for observed over expected scores (fitdistrplus R package), which enables selection of the most differential contacts located within the expected 5% and 95% tails of the fitted distributions. We also obtained the contacts with strongest differences in their ranks by sorting all contacts based on their value intensity and selecting the top and bottom 5% based on their rank difference from each genomic distance.

We decided to define Hi-C-specific and GAM-specific contacts by the z-score approach, given that the 5% result sets from all transformations yielded comparable set sizes with a high mutual overlap (~300,000, Extended Data Fig. 7), while the z-score-derived sets were the least affected by under-detected regions and accounted for the observed decay of mean contact frequency over distance (Extended Data Fig. 7). In addition to the 5% Hi-C-specific and GAM-specific contact sets with 231,164 and 265,166 contacts, respectively, we also extracted two contact sets from 10% tails with 473,884 and 499,198 contacts, respectively (Supplementary Table 7).

In addition to identifying the most differential contacts, we defined a set of strong-and-common contacts that differ very little in value intensities between GAM and Hi-C. We first ranked all contacts with a delta z-score of less than 1.0 according to the lower z-score value from GAM and Hi-C, and extracted the strongest 5% or 10% of contacts for each chromosome (total of 148,536 and 297,064 contacts, respectively).

We used the definition of significant Dixon Hi-C contacts published by Fit-Hi-C[46]. We selected all *cis* contacts from the pre-processed data (two-pass spline interpolating on 10 consecutive restriction enzyme fragments cut by NcoI) with genomic separation below 4 Mb and a *q* value < 0.05. Next, we assigned the contacts into 40 kb windows based on their fragment midpoints.

## Feature enrichments within differential contacts

We queried whether contacts identified to be specific for GAM and Hi-C are associated with specific biological features (Extended Data Fig. 8a) relative to those obtained from randomized data. First, we produced three permutations for each of the foreground sets by random sampling the same number of contacts with the same genomic distance out of all contacts of the same chromosome not contained in the respective foreground set. Next, we established a feature table listing the presence or absence of 14 selected features in 40 kb windows (Supplementary Table 6) and checked for the pairwise presence of 105 homotypic and heterotypic feature combinations in the subsets of Hi-C-specific, GAM-specific and strong-and-common contacts. Here, we annotated 98,600 (42%), 164,946 (62%) and 78,919 (53%) of 5% contact sets with the presence of any feature pair, respectively (Supplementary Table 7).

To determine which feature combinations are amplified at contact anchor points, we computed the relative occurrence (frequency of feature pair <i,j> in total contact set) for each feature pair in the contact set. We ranked the results by descending Gini impurity obtained using the random forest classification, which was trained to distinguish the annotated GAM-specific and Hi-C-specific contacts based on the presence of associated feature pairs (sklearn 0.19.2, 500 trees with fivefold cross validation, max_features as sqrt(num_features), criterion = 'gini', no max depth). For further investigation we selected the top 10 feature pairs most informative for binary classification, omitting enriched feature pairs of lower genomic abundance. Given that feature pairs with higher frequency in Hi-C than in GAM are not part of this subset, we added the top three feature pairs showing the strongest amplification in Hi-C relative to GAM (Supplementary Table 8).

Different features can often be found to be co-present at the same anchor points of a contact. We applied the UpSetR package[47] to the 5% sets of contacts from GAM-specific, Hi-C-specific, strong-and-common, and the genome-wide background. We plotted the abundance of a feature pair according to the percentile of feature occurrence, along with the number of observed co-localization events between pairs of features. We established the genome-wide background set by randomly selecting 5% of all non-zero contacts observed by GAM and Hi-C.

## Analysis of GAM-preferred and Hi-C-preferred regions

We assessed the preference towards contributing to GAM-specific contacts or Hi-C-specific contacts for each genomic 40 kb window.

First, we counted how often a window was an anchor point for contacts of the GAM-specific or Hi-C-specific subsets. Next, we calculated the absolute difference between both counts and estimated a 90% percentile cut-off for each chromosome. We considered genomic windows above this threshold to hold either mostly GAM-specific contacts or mostly Hi-C-specific contacts. In total, this resulted in 6,520 windows being labeled as GAM-preferred regions by having predominantly GAM-specific contacts, and 5,926 as Hi-C-preferred regions with a much higher count of Hi-C-specific contacts over GAM-specific contacts. Similarly, we identified genomic regions that are equally well detected by GAM and Hi-C (common). Here, for each genomic window we counted the number of anchor points from contacts of the strong-and-common set and selected the top 10% genomic windows with the highest counts from each chromosome.

Next, we assessed gene density and transcriptional activity in groups of genomic regions using published gene annotations and mESC-46C TPM values[48]. We transferred the provided mm10 gene positions to mm9 using UCSC liftover[49] and assigned genes to genomic windows of 40 kb using bedtools intersect. We annotated lamina associations within 40 kb genomic regions according to mESC LaminB1 HMM calls[10]. The genome-wide LAD ratio was computed as the number of positive HMM state calls over the total number of windows. For markers of transcription factors, histone modifications and RNA polymerase II states, we used 40 kb window classification for peak and feature presence (Supplementary Table 5) and counted the number of positive windows in each subset.

## Analysis of interaction complexity

We used SLICE to compute the three-way probability of interaction ($Pi_{ABC}$) and identified 1 Mb intrachromosomal triplets from the GAM-1,250 dataset where $Pi_{ABC} < 0.05$ (Supplementary Note). In this work, we define complexity as the mean number of triplets with $Pi_{ABC} < 0.05$ over all combinations of B and C windows for a given A window (where complexity is calculated for a genomic region, Fig. 5b,c), or the mean over all C windows for a given A and B window (where complexity is calculated for a pairwise contact, Fig. 5d,e).

For each genomic window labeled as a Hi-C preferred region, common region or GAM-preferred region, we checked for the compartment assignment and correlated the outcome with the complexity at the 1 Mb genomic window. Similarly, we estimated the complexity of genomic and epigenetic features by categorizing 40 kb genomic windows according to the presence of transcription factors, histone modifications and RNA polymerase II states, and presenting the complexity of the respective 1 Mb genomic window.

We identified potential Hi-C triplets using matrices of normalized ligation frequencies at 1 Mb binning[6]. On the basis that if a triplet (ABC) is formed, the three component pairwise interactions (AB, AC, BC) should all be detected by Hi-C, we therefore estimated Hi-C triplet intensity as the minimum ligation frequency of the three component pairwise interactions making up the triplet. We then selected the strongest 2% of all Hi-C triplets from every genomic distance for which there were at least 500 possible triplets.

To identify triplet contacts from single-cell Hi-C data, we downloaded 10 haplotype-resolved 3D models of chromatin folding in single cells generated by Dip-C (GSE117109)[12] analysis of diploid mouse embryonic stem cells kept in serum media (GSE94489)[7]. We used the bedtools window to generate a list of 1 Mb bins from mm9, used liftOver to convert each bin to mm10 coordinates and calculated the 3D position as the centroid of all overlapping 10 kb bins from the modeling data. For each chromosome, we examined 20 structures (maternal and paternal chromosomes for each of 10 cells). We reasoned that if three loci form a triplet in single cells, then the pairwise distances between the three loci should all be small. We therefore scored every possible triplet by calculating the maximum of the three pairwise distances (AB, AC, BC) in each model individually and then taking the minimum score across

all 20 models. We selected the best triplets as the lowest 2% from every genomic distance for which there were at least 500 possible triplets.

## Reporting summary

Further information on research design is available in the Nature Portfolio Reporting Summary linked to this article.

## Data availability

GAM sequencing data generated for this study are available from GEO (GSE166381). The original GAM sequencing data[3] are available as a separate accession (GSE64881). Other datasets used in the study are listed in Supplementary Table 5, and additional intermediate data are available on GitHub (https://github.com/pombo-lab/multiplex-gam-2023/tree/main/data).

## Code availability

GAM sequencing samples were processed using GAMtools v1.1.0, which is available at https://github.com/pombo-lab/gamtools/releases/tag/v1.1.0. Custom code used for data analysis is available at https://github.com/pombo-lab/multiplex-gam-2023.

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

## Acknowledgements

The authors thank all laboratory members and the 4D Nucleome consortium for helpful discussions. The authors also thank W. Winick-Ng for providing illustrations in Fig. 1c,d and Extended Data Fig. 2, and deeply appreciate the feedback from T. Szczepińska on the processing of differential contacts. A.P. acknowledges support from the Helmholtz Association (Germany) and from the Deutsche Forschungsgemeinschaft (DFG; German Research Foundation) under Germany's Excellence Strategy - EXC-2049 - 390688087. A.P. and M.N. acknowledge support from the National Institutes of Health Common Fund 4D Nucleome Program grant U54DK107977, and the Berlin Institute of Health (BIH). M.N. acknowledges the support of the CINECA ISCRA Grant HP10CYFPS5 and HP10CRTY8P, computer resources at INFN and Scope at the University of Naples (M.N.). L.R.W. acknowledges the support of Ohio University's GERB program. R.A.B. is supported by the Wellcome Trust (209181/Z/17/Z and 224135/Z/21/Z). Some computational work was supported by the Wellcome Trust Core Award Grant Number 203141/Z/16/Z with additional support from the NIHR Oxford BRC.

## Author contributions

A.P. and M.N. devised the multiplex-GAM strategy; R.A.B., A.K. and R.K. produced GAM datasets; R.A.B., A.K. and R.K. optimized the experimental protocol; R.A.B. performed GAM data processing, quality control and in silico merging experiments; C.A., A.S., S.B., A.M.C. and M.N. developed and implemented the updated SLICE model; C.A. and A.S. applied SLICE to determine the optimal experimental parameters and performed SLICE analysis of GAM data; R.A.B. performed enrichment analyses for SLICE contact pairs and triplets; C.J.T. and M.S. performed TAD boundary analysis; C.J.T. and A.K. processed epigenetic and genomic data; C.J.T. and M.S. devised the pipeline to extract differential contacts; C.J.T. generated randomized contact sets; Y.Z. and Y.L. developed and applied the pipeline for feature pair presence and enrichments over background; C.B. performed the random forest discrimination tests; C.B. and C.J.T. performed feature pair co-localization tests; C.J.T. identified and analyzed preferred regions; R.A.B. and C.J.T. assessed complexity of contacts, preferred regions and feature-positive genomic windows; T.D., R.A.B. and C.J.T. performed TAD calling of sc-Hi-C data; R.A.B. performed triplet analysis of sc-Hi-C models; A.P. supervised GAM experiments and bioinformatics analyses; L.R.W. supervised feature enrichment analyses in specific contacts; M.N. supervised SLICE development; A.P., R.A.B., C.J.T., A.K., C.A., A.S., Y.Z., C.B., L.R.W. and M.N. contributed to the interpretation of the results; R.A.B. wrote the first draft of the manuscript; R.A.B. and C.J.T. designed the figures; R.A.B., C.J.T. and A.P. wrote the manuscript; and all authors provided critical feedback and helped revise the manuscript.

## Funding

## Competing interests

A.P., M.N., R.A.B and A.S. hold a patent on 'Genome Architecture Mapping': Pombo, A., Edwards, P. A. W., Nicodemi, M., Scialdone, A., Beagrie, R. A. Patent PCT/EP2015/079413 (2015). All other authors have no competing interests.

## Additional information

**Extended data** are available for this paper at https://doi.org/10.1038/s41592-023-01903-1.

**Correspondence and requests for materials** should be addressed to Lonnie R. Welch, Mario Nicodemi or Ana Pombo.

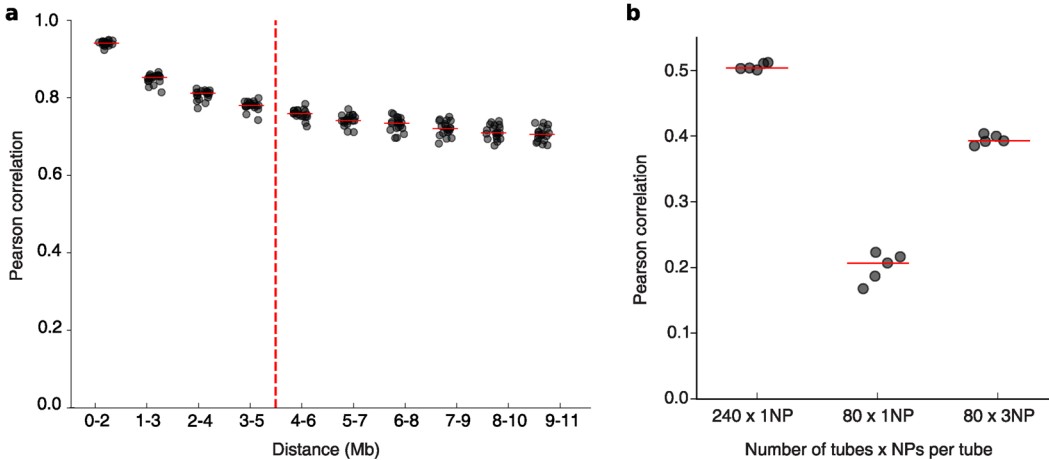

**Extended Data Fig. 1 | In silico validation of multiplexing GAM. a**, Correlation of GAM datasets containing single nuclear profiles with GAM datasets where the same NPs are merged *in silico* in sets of three, stratified by distance. **b**, Correlation between two GAM datasets of 240 x 1NP libraries (left) compared to the correlation of two datasets of 80 x 1NP libraries (middle) and two replicates of 80 x 3NP (right, total number of NPs is 240, same as left bar, and same sequencing cost as middle bar). Horizontal lines show mean of five randomly sampled subsets.

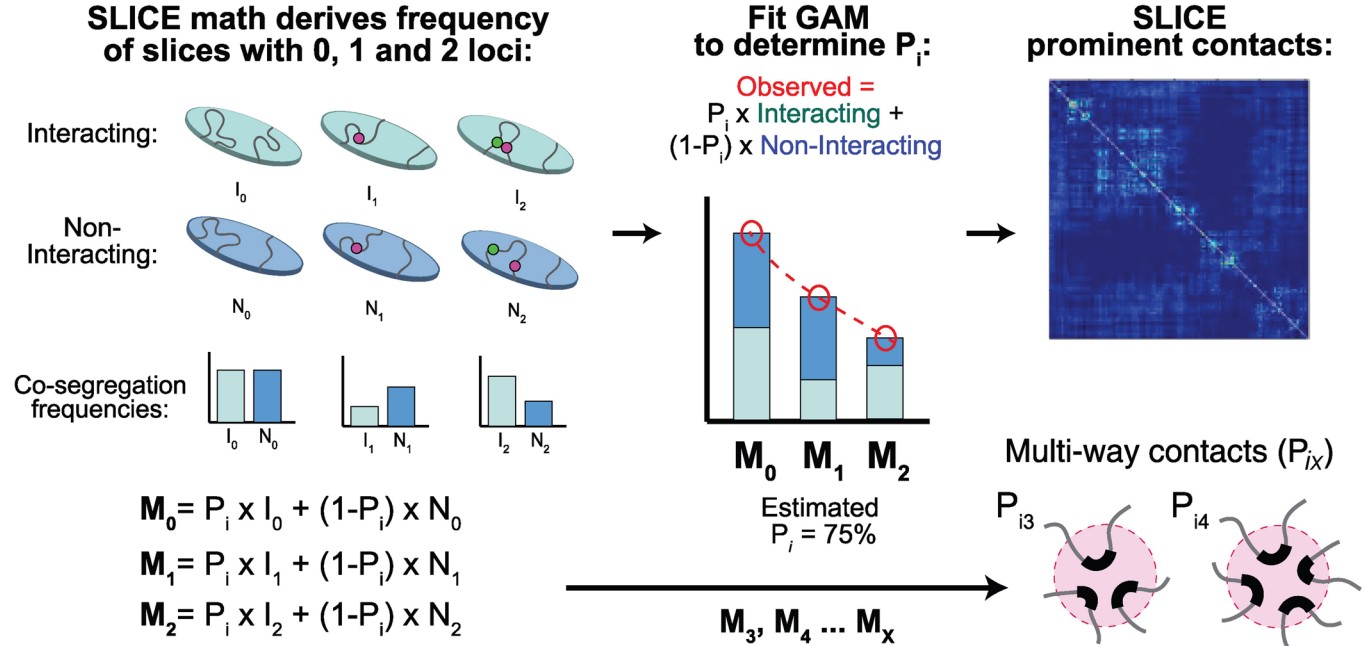

**a**

## Locus State

### Cell population: 75% interaction frequency

Interacting (I)     Non-Interacting (N)

Frequency = Probability of interaction, $P_i$     Frequency = $1 - P_i$

**b**

## Locus Co-Segregation in GAM

Neither locus present ($M_0$):     One locus present ($M_1$):     Both loci present ($M_2$):

Compute co-segregation frequency

↓

GAM D' matrix, co-segregation matrix,...     →     TAD calling, compartment calling, ...

**c**

## SLICE Modelling

### SLICE math derives frequency of slices with 0, 1 and 2 loci:

Interacting:     $I_0$     $I_1$     $I_2$

Non-Interacting:     $N_0$     $N_1$     $N_2$

Co-segregation frequencies:     $I_0$  $N_0$     $I_1$  $N_1$     $I_2$  $N_2$

$M_0 = P_i \times I_0 + (1-P_i) \times N_0$

$M_1 = P_i \times I_1 + (1-P_i) \times N_1$

$M_2 = P_i \times I_2 + (1-P_i) \times N_2$

### Fit GAM to determine $P_i$:

Observed =
$P_i \times$ Interacting +
$(1-P_i) \times$ Non-Interacting

$M_0$  $M_1$  $M_2$

Estimated $P_i = 75\%$

### SLICE prominent contacts:

Multi-way contacts ($P_{ix}$)

$P_{i3}$     $P_{i4}$

$M_3, M_4 ... M_x$

**Extended Data Fig. 2 | Concept of the SLICE model. a**, Assuming a cell population with an actual interaction frequency of 75%, nuclear cryosections intersecting both loci can be obtained from nuclei in both interacting and non-interacting states. **b**, For any pair of loci, a GAM cryosection might detect either both loci, one locus, or neither. Co-segregation frequency or normalized linkage (D') can be applied to infer proximity of the two loci. **c**, SLICE provides a statistical model to identify pairs of loci interacting above the background at their genomic distance, and to estimate the proportion of cells in the population with pairs in the interacting state (Probability of interaction or Pi).

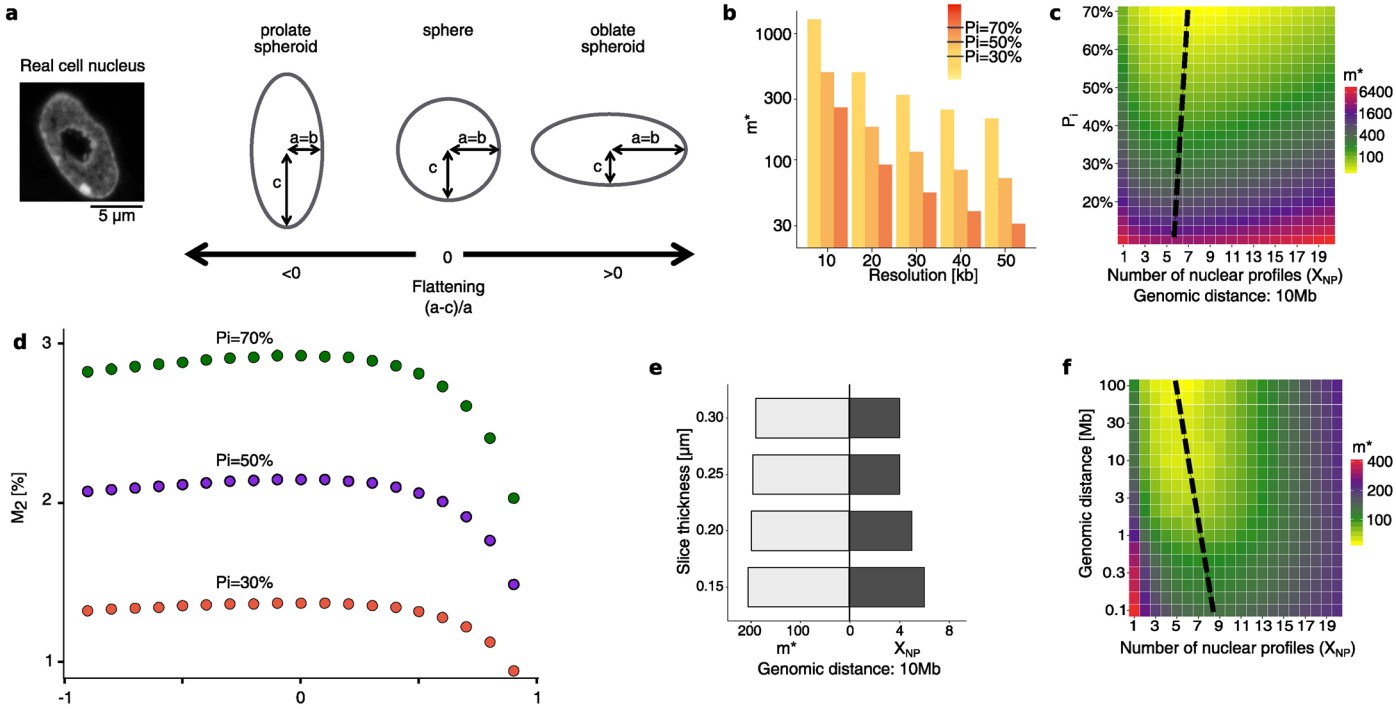

**Extended Data Fig. 3 | Extending the SLICE model parameters to include additional metrics. a**, SLICE can be generalized to account for complex nuclear shapes. As a case study, we considered spheroids (that is, ellipsoids with two equal axes), which we parametrized by the flattening parameter (the relative difference of the radii). **b**, Relationship between the minimum number of tubes required ($m^*$), genomic resolution and sensitivity. **c**, $m^*$ as a function of $X_{NP}$ and the interaction probability ($Pi$) given perfect detection efficiency, a genomic distance of 10Mb and $h$=220 nm, at 30 kb resolution. Dashed black line marks the approximate position of the minimum value of $m^*$ across each row. **d**,

Co-segregation frequency of pairs of loci ($M2$) derived by SLICE as function of the flattening parameter in spheroidal nuclei of constant volume, for different values of interaction frequency ($Pi$) given slice thickness ($h$) of 220 nm, resolution of 30 kb, detection efficiency of 1 and genomic distances of 50 Mb. **e**, Optimal value of $X_{NP}$ and the corresponding value of $m^*$ for a range of possible slice thicknesses given $Pi$ =30%, genomic distance of 10Mb and perfect detection. **f**, Required number of tubes ($m^*$) as a function of $X_{NP}$ and the genomic distance given perfect detection, $Pi$=50% and $h$=220 nm, at 30 kb resolution. Dashed black line marks the position of the minimum value of $m^*$ across each row.

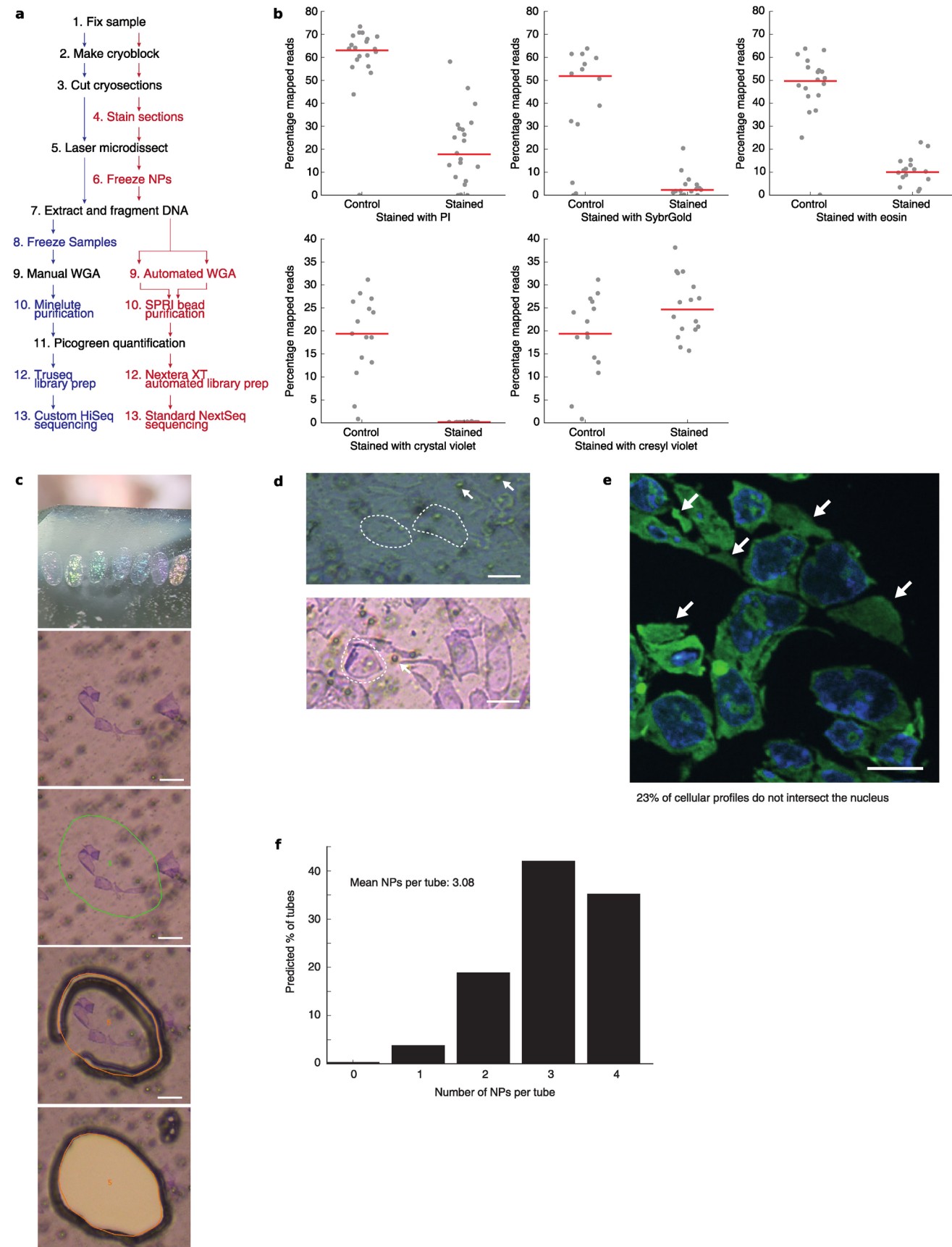

**Extended Data Fig. 4 | See next page for caption.**

**Extended Data Fig. 4 | Optimizations to the GAM protocol. a**, Comparison of the original (black and blue lettering) and multiplex (black and red lettering) GAM protocols. WGA: Whole Genome Amplification. **b**, Effect of various staining protocols on downstream DNA extraction. Top row: propidium iodide (n=21 each), SYBR® Gold (n=16 stained; 14 unstained) and Eosin (n=16 stained; 19 unstained); bottom row: crystal violet (n=16 stained; 15 unstained) and cresyl violet (n=16 stained; unstained the same as for crystal violet). Red lines indicate the median percentage of mapped reads per GAM-3NP sample. **c**, Top panel: cryosection thickness can be identified from the section's color. Bottom panels: cresyl violet staining improves the visualization of NPs during laser microdissection. Scale bar 10μm. **d**, Top: unstained cryosections from mES cells. Individual NPs are outlined by dashed white lines. White arrows indicate

typical background (air bubbles) in the microdissection membrane. Bottom: Cresyl violet stains both cytoplasm and nucleoplasm in mES cell cryosections as visualized by brightfield microscopy, and therefore does not distinguish cellular profiles that intersect the nucleus. Scale bar 10μm. **e**, Example mES cell cryosection visualized by confocal fluorescence microscopy. DNA stained with DAPI is shown in blue and RNA stained with SYTO® RNASelect dye is shown in green. White arrows indicate cellular profiles that do not intersect nuclei (n=647 profiles intersect the nucleus of 857 profiles analyzed; Supplementary Table 10). Scale bar 10μm. **f**, Binomial distribution showing the expected number of profiles per GAM sample that intersect nuclei across a collection of samples each with four cellular profiles.

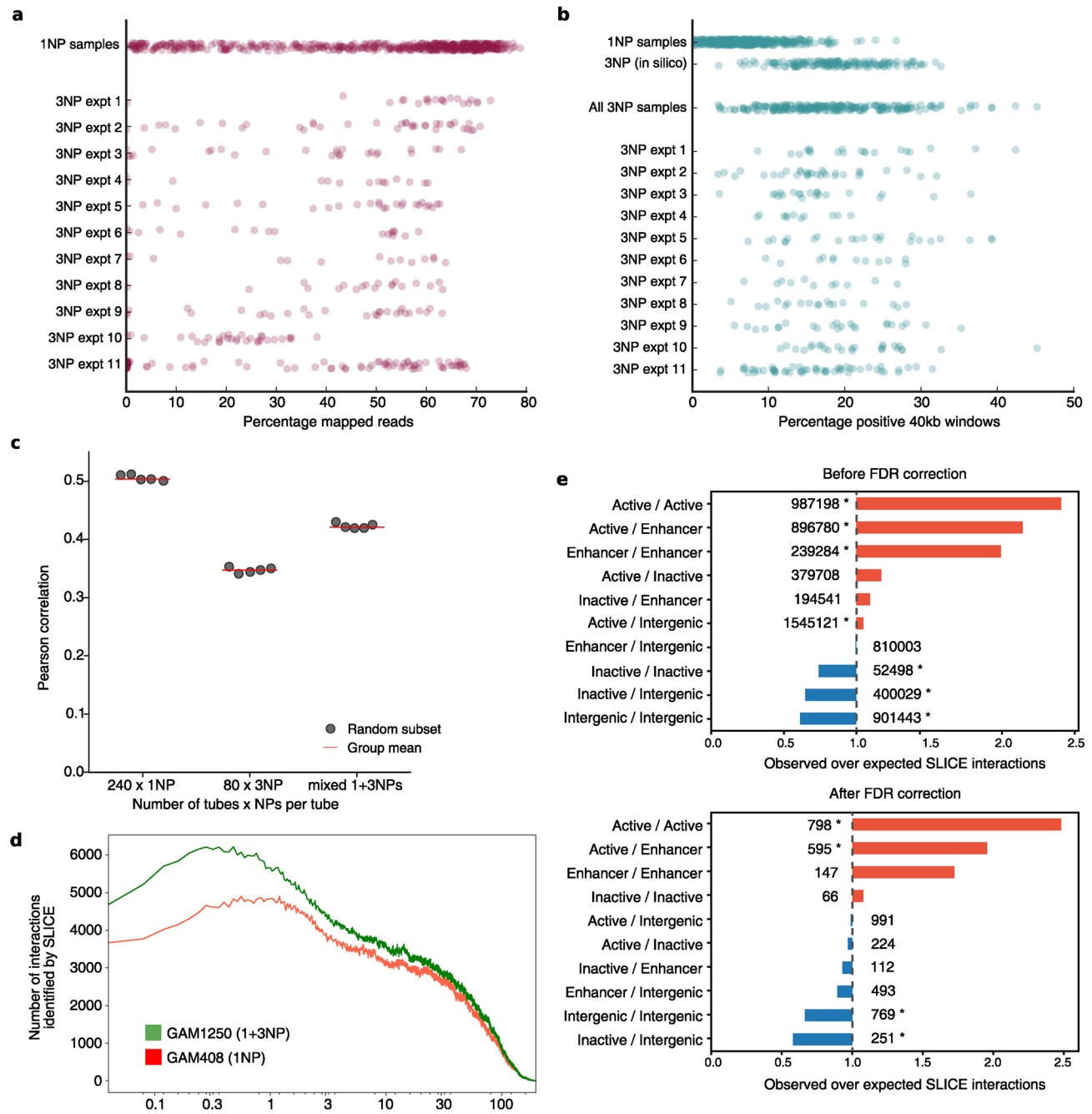

**Extended Data Fig. 5 | Comparison of multiplex-GAM to single-NP GAM. a,** Percentage of reads mapping to the mouse genome for 1NP libraries and for 3NP libraries separated by experimental batch. **b,** Percentage of positive 40 kb windows for 1NP libraries, for 1NP libraries combined into 3NP libraries *in silico* and for experimental 3NP libraries (that is four cellular profiles dissected into each tube with binomial expectation that, on average, three profiles in each tube intersect the nucleus; see Extended Data Fig. 4e,f) separated by experimental batch. **c,** Correlation between two GAM datasets of 240 x 1NP libraries (left) compared to the correlation of two replicates of 80 x 3NP (middle) and a mixture of 96 x 1NP and 48 x 3NP (right, same number of total NPs mixed in the same ratio

as the GAM-1250 dataset). Left and middle bars are the same as in Extended Data Fig. 1b. Horizontal line indicates mean over five randomly sampled subsets. **d,** Number of prominent interactions identified by SLICE at each genomic distance in the merged GAM-1250 dataset and the published mES GAM-408 dataset[3]. **e,** Enrichment analysis of pairwise interactions identified by SLICE involving active, inactive, intergenic or enhancer regions for the merged GAM-1250 dataset either before (top) or after (bottom) false discovery rate correction. Statistically significant enrichments/depletions (those falling outside 95% of randomized observations after Bonferroni correction) are marked by an asterisk.

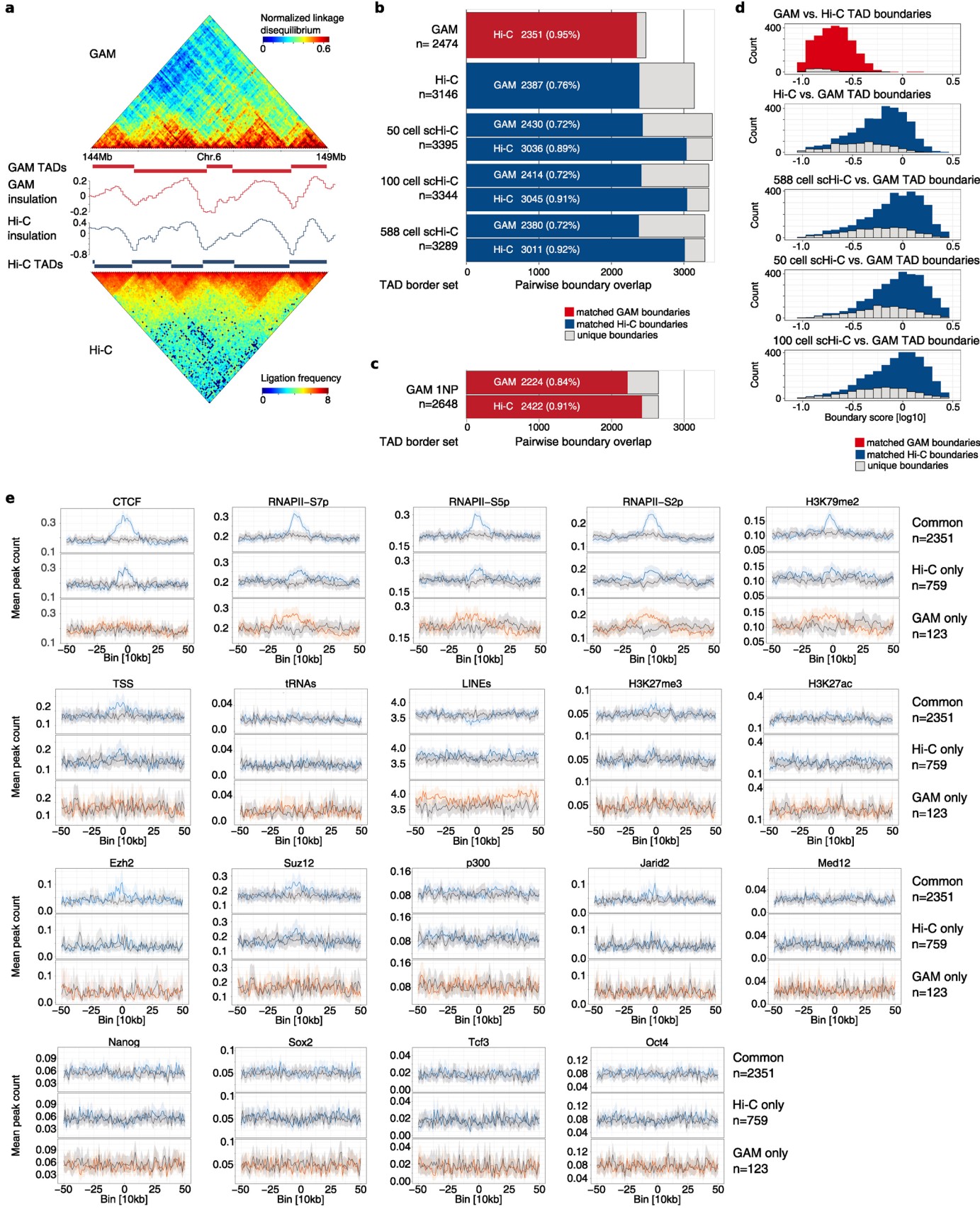

**Extended Data Fig. 6 | See next page for caption.**

**Extended Data Fig. 6 | Similarity at TAD level for GAM, bulk Hi-C and sc-Hi-C. a**, Comparison of topological associated domains (TADs) defined by the insulation score method in GAM (top) and Hi-C[6] (bottom). **b**, Overlap of TAD boundaries detected in GAM-1250 (red) and Hi-C (blue) as well as boundaries detected in sc-Hi-C contact maps produced from 50 cells, 100 cells, and all 588 cells[7]. **c**, Overlap between TAD boundary calls from 1NP GAM data with Hi-C data and full 1+3NP GAM-1250 data. **d**, Boundary strength (drop of insulation) measured for GAM (red), Hi-C (blue) and sc-Hi-C contact maps produced from 50 cells, 100 cells, and all 588 cells was calculated for TAD boundaries which are shared between the sets (dark bars) or only found in a single set (light bars). **e**, Mean frequency and 95% bootstrap CI (shaded) of RNA-Pol II, transcription factor occupancy and epigenetic marks centered at TAD boundary sites. Common, Hi-C-specific, GAM-specific boundaries are shown in black, dark blue, and red, respectively, gray values are based on shuffled boundary positions of each category.

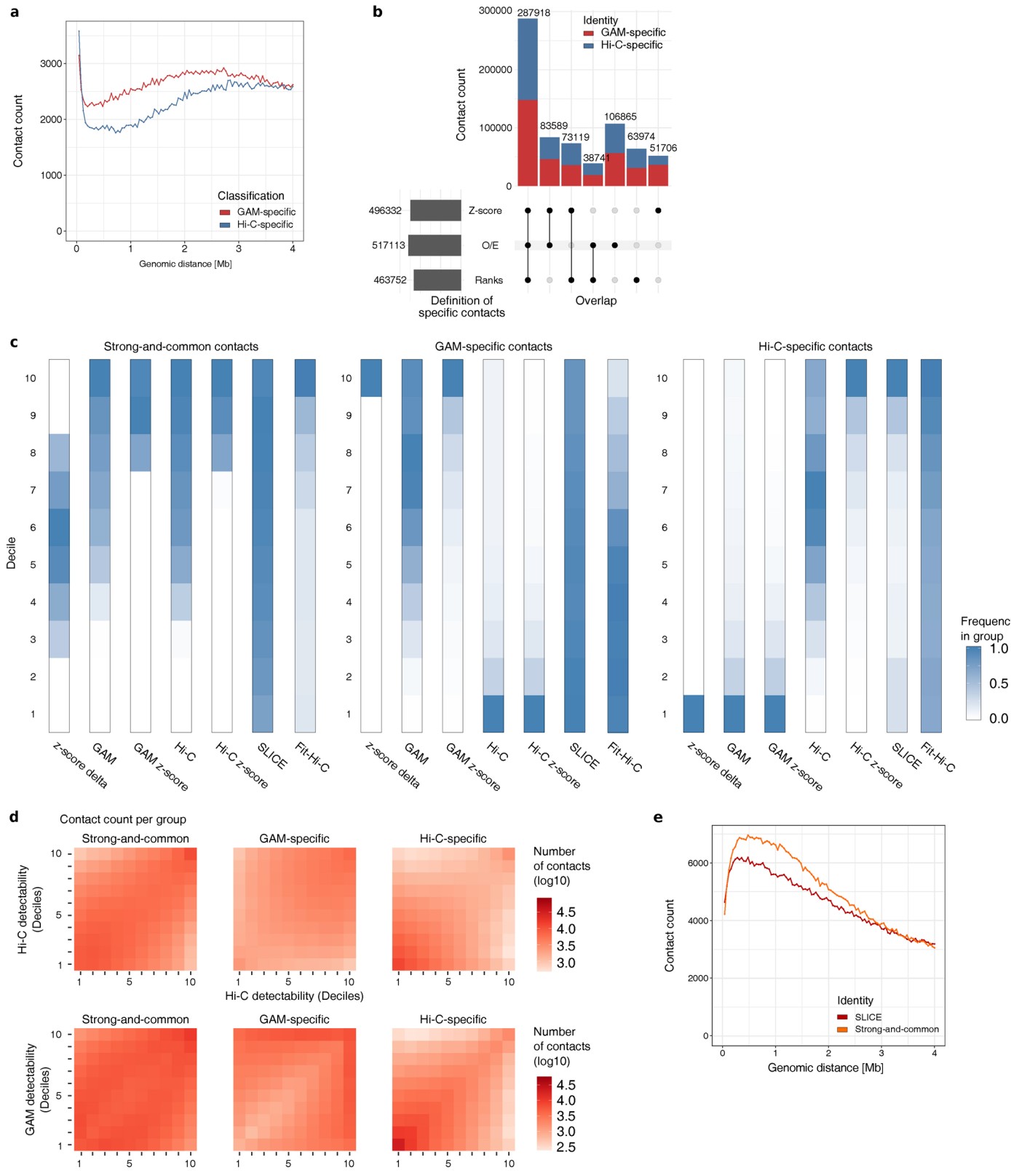

**Extended Data Fig. 7 | See next page for caption.**

**Extended Data Fig. 7 | Intrinsic properties of method-specific contact subsets and contacts common between GAM and Hi-C. a**, Distribution by genomic distance of GAM-specific (red) and Hi-C-specific (blue) contacts. **b**, Upset plot presenting the agreement between sets of differential contacts obtained by different data transformations (Z-score transformation; Observed over expected, O/E; Rank transformation). The fraction of contacts in the intersections designated as GAM-specific or Hi-C-specific is colored in red and blue, respectively. **c**, Decile distributions of contact intensities before and after z-score transformation for the sets of GAM-specific contacts, Hi-C-specific

contacts, and Strong-and-common contacts. **d**, Detectability of GAM-specific contacts, Hi-C-specific contacts, and Strong-and-common contacts. Windows are split into decile groups based on their detectability. Heatmaps show the log-scaled number of contacts connecting windows in different detectability deciles. Top: Deciles calculated by Hi-C detectability (total number of ligation events per window). Bottom: Deciles calculated by GAM detectability (window detection frequency). **e**, Distribution by genomic distance of SLICE interactions (red) and strong-and-common contacts (orange).

**a**

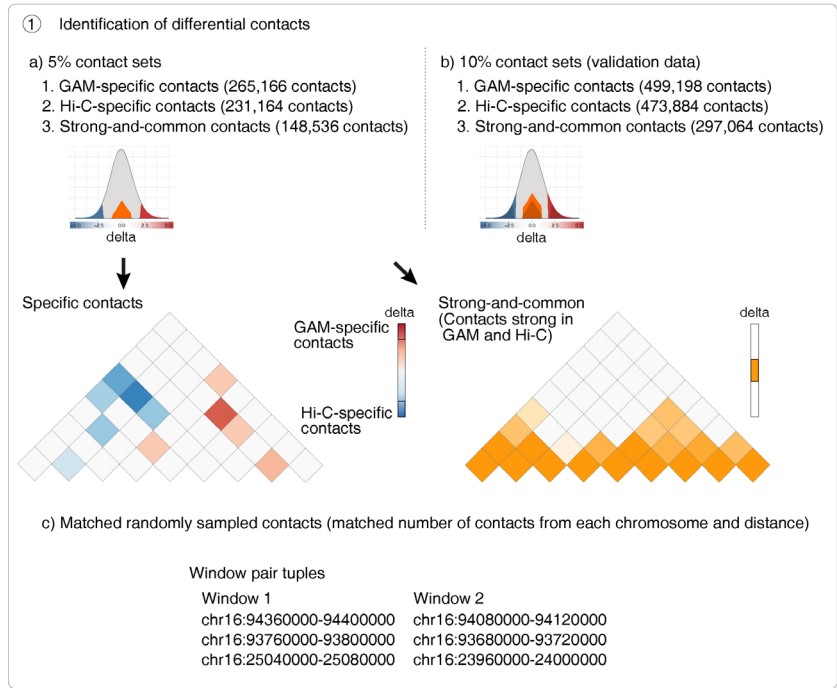

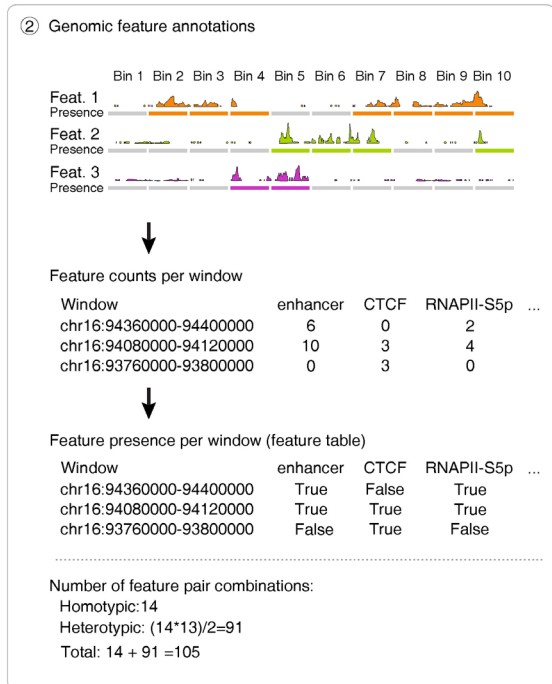

③ Feature annotation of contact window pairs

   Case 1: Feature A occurs in Window 1 and Feature B occurs in Window 2
   Case 2: Feature A occurs in Window 2 and Feature B occurs in Window 1

④ Select contacts with annotated feature pairs (Fig. 3a)

⑤ Calculate amplification and discriminatory power of feature pairs (Fig. 3b; Suppl. Fig. 8b)

   - Statistics table
   - Relative presence of feature pairs in contact sets

⑥ Discrimination between GAM and Hi-C (Fig. 3b,c; Suppl. Fig. 8c)

   - Selection of most descriptive feature pairs for discriminating GAM-specific contacts from Hi-C-specific contacts (Random Forest)
   - Most amplified feature pairs in GAM-specific contacts (vs. Hi-C-specific contacts)
   - Most amplified feature pairs in Hi-C-specific contacts (vs. GAM-specific contacts)

**b**

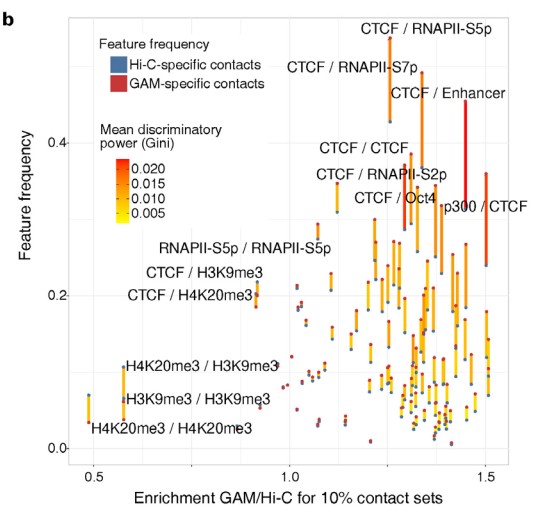

**c**

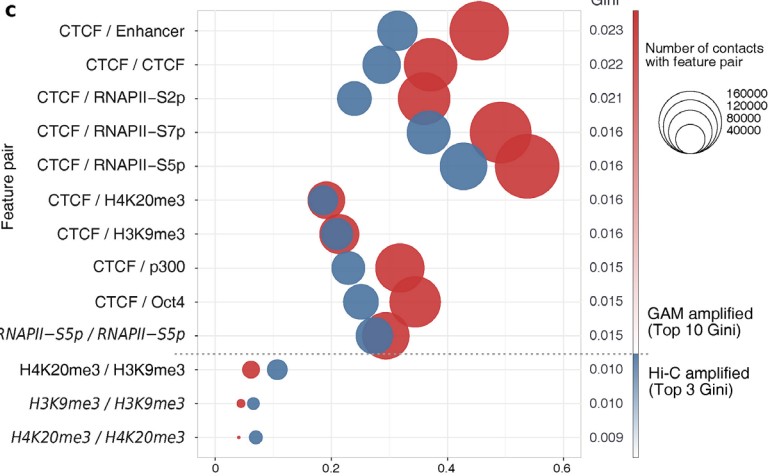

**Extended Data Fig. 8 | Annotation pipeline for feature pair presence and evaluation of sets with 10% of strongest contacts. a**, Strategy for detecting enrichment of feature pairs within sets of specific contacts. **b**, Pairwise differences of feature pair frequencies observed in the 10% sets of GAM-specific and Hi-C-specific contacts relative to their relative enrichment. Differences are colored according to the Gini impurity obtained by Random Forest predictions. **c**, Feature pairs with the highest discriminatory power in the 10% subsets relative to the abundance of the feature pair in the contact sets. Pairs with different order relative to the 5% results set italic.

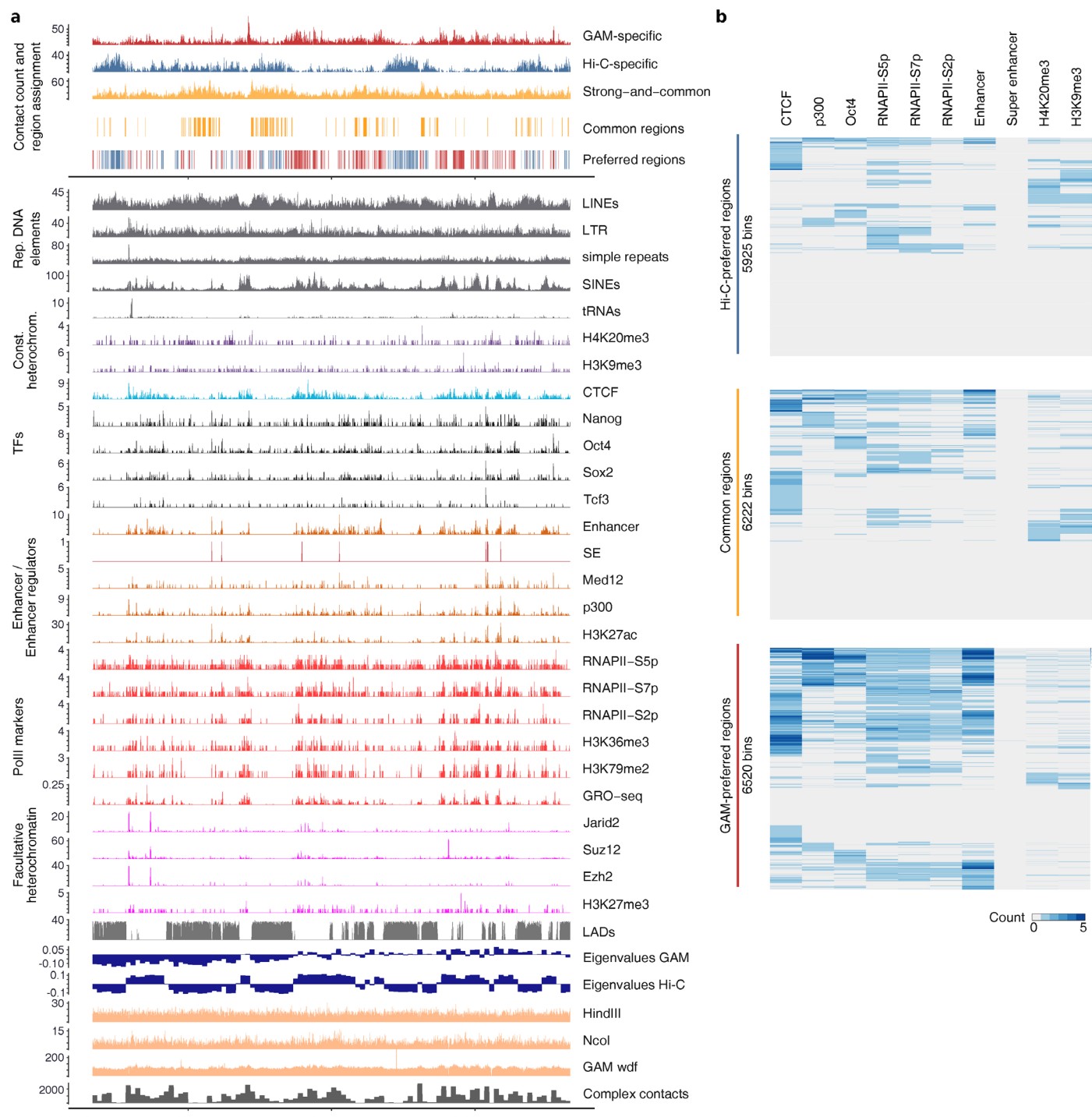

**Extended Data Fig. 9 | Feature presence at genomic windows with increased number of method-specific contacts. a**, Feature tracks and assignment of genomic regions having mostly GAM-specific contacts (GAM-preferred regions) or Hi-C-specific contacts (Hi-C-preferred regions). **b**, Heatmap representation for feature presence at GAM-preferred regions, Hi-C-preferred regions) or regions with contacts found at similar intensity by both methods (common regions).

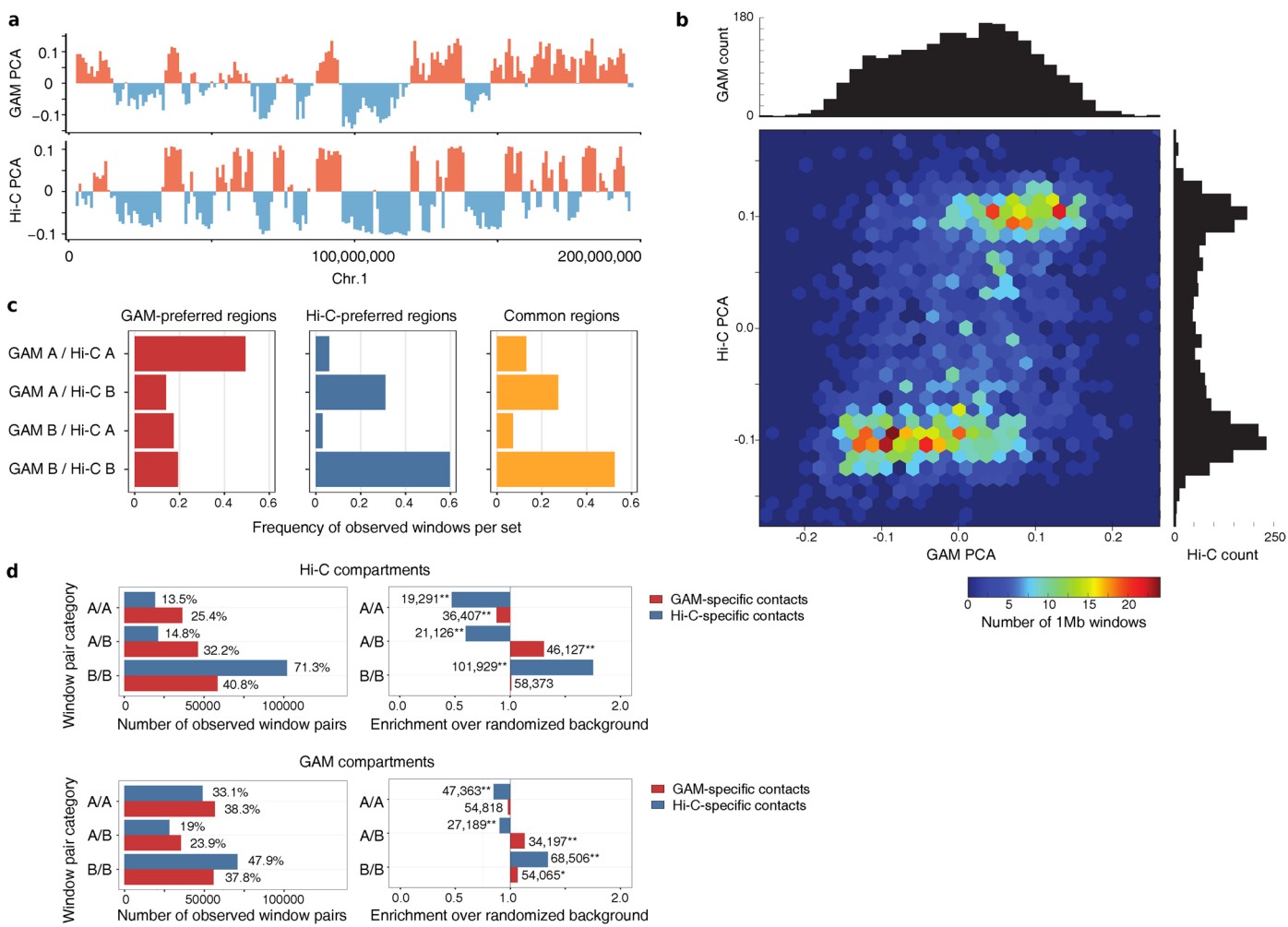

**Extended Data Fig. 10 | Compartment-association of GAM-specific contacts.**
**a**, PCA eigenvalues from GAM and Hi-C for mouse chromosome 1. **b**, Correlation heatmap for GAM and Hi-C eigenvalues. A small subset of windows assigned to the A compartment by GAM (positive eigenvalues) are assigned to the B compartment by Hi-C (negative eigenvalues). Value histograms indicate that Hi-C eigenvalues are clearly bimodal distributed while GAM finds an increased number of eigenvalues in the transition zone. **c**, Agreement between GAM and Hi-C compartment calls for GAM-preferred (left), Hi-C preferred (middle) and common (right) regions. **d**, Absolute number (left) and enrichment over random background (right) of compartment combinations observed in the sets of Hi-C-specific contacts (blue) and GAM-specific contacts (red). Compartment definitions of 1Mb resolution were obtained from Hi-C data (top) or GAM data (bottom); * P < 0.05, ** P < 0.01, empirical test, n = 500 contact permutations.

| | |
|---|---|

# Reporting Summary

Nature Research wishes to improve the reproducibility of the work that we publish. This form provides structure for consistency and transparency in reporting. For further information on Nature Research policies, see our Editorial Policies and the Editorial Policy Checklist.

## Statistics

For all statistical analyses, confirm that the following items are present in the figure legend, table legend, main text, or Methods section.

| n/a | Confirmed | |
|---|---|---|
| ☐ | ☒ | The exact sample size (*n*) for each experimental group/condition, given as a discrete number and unit of measurement |
| ☐ | ☒ | A statement on whether measurements were taken from distinct samples or whether the same sample was measured repeatedly |
| ☐ | ☒ | The statistical test(s) used AND whether they are one- or two-sided <br> *Only common tests should be described solely by name; describe more complex techniques in the Methods section.* |
| ☒ | ☐ | A description of all covariates tested |
| ☐ | ☒ | A description of any assumptions or corrections, such as tests of normality and adjustment for multiple comparisons |
| ☐ | ☒ | A full description of the statistical parameters including central tendency (e.g. means) or other basic estimates (e.g. regression coefficient) AND variation (e.g. standard deviation) or associated estimates of uncertainty (e.g. confidence intervals) |
| ☒ | ☐ | For null hypothesis testing, the test statistic (e.g. *F*, *t*, *r*) with confidence intervals, effect sizes, degrees of freedom and *P* value noted <br> *Give P values as exact values whenever suitable.* |
| ☒ | ☐ | For Bayesian analysis, information on the choice of priors and Markov chain Monte Carlo settings |
| ☒ | ☐ | For hierarchical and complex designs, identification of the appropriate level for tests and full reporting of outcomes |
| ☐ | ☒ | Estimates of effect sizes (e.g. Cohen's *d*, Pearson's *r*), indicating how they were calculated |

*Our web collection on statistics for biologists contains articles on many of the points above.*

## Software and code

Policy information about availability of computer code

| Data collection | No software was used for the collection of data in this study |
|---|---|
| Data analysis | GAM sequencing data was analysed using GAMtools v1.1.0, samtools v0.9, bedtools v2.27.1 and bowtie2 v2.1.0. Peaks in ChIP-seq data were called using BCP v1.1. Additional custom Python v3.7 scripts used for data analysis are available at https://github.com/pombo-lab/multiplex-gam-2022 |

For manuscripts utilizing custom algorithms or software that are central to the research but not yet described in published literature, software must be made available to editors and reviewers. We strongly encourage code deposition in a community repository (e.g. GitHub). See the Nature Research guidelines for submitting code & software for further information.

## Data

Policy information about availability of data

All manuscripts must include a data availability statement. This statement should provide the following information, where applicable:
- Accession codes, unique identifiers, or web links for publicly available datasets
- A list of figures that have associated raw data
- A description of any restrictions on data availability

The original GAM sequencing data (Beagrie et al., 2017) are available from GEO (GSE64881). GAM sequencing data generated for this study are available as a separate accession (GSE166381). Hi-C data used are available from GEO (GSE35156). Other datasets (listed in Supplementary Table 4) are H3K9me3 (GSE18371), H4K20me3 (GSE12241), CTCF (GSE29184), Nanog (GSE11724), Oct4 (GSE11724), Sox2 (GSE11724), Tcf3 (GSE11724), Med12 (GSE22557), p300 (GSE29184), H3K27ac (GSE29184), GRO-seq (GSE27037), RNAPII-S7p (GSE94364), RNAPII-S5p (GSE94364), RNAPII-S2p (GSE34520), Total RNAPII (8GW16) (GSE94364), H3K36me3 (GSE11724), H3K79me2 (GSE11724), Jarid2 (GSE18776), Suz12 (GSE18776), Ezh2 (GSE18776), H3K27me3 (GSE94364), mESC HMM state calls (GSE17051)

# Field-specific reporting

Please select the one below that is the best fit for your research. If you are not sure, read the appropriate sections before making your selection.

☒ Life sciences          ☐ Behavioural & social sciences          ☐ Ecological, evolutionary & environmental sciences

For a reference copy of the document with all sections, see nature.com/documents/nr-reporting-summary-flat.pdf

# Life sciences study design

All studies must disclose on these points even when the disclosure is negative.

| | |
|---|---|
| Sample size | The new GAM dataset was deemed sufficiently large because 80% of 40kb windows were detected at least 40 times. |
| Data exclusions | GAM datasets with <15% mapped reads were excluded from further analysis (Beagrie et al. 2017), as were all samples in experimental batches that were deemed to have failed (e.g. those stained with SybrGold, Extended Fig 4b). All GAM samples (including all excluded samples) are outlined in Supplementary Table 2 and all sequencing data is available under GEO accession GSE166381. |
| Replication | GAM datasets were derived from two biological replicates of mouse ES cells, independently grown at different times. Details of which samples belong to each replicate can be found in Supplementary Table 2. No significant difference was found between the two biological groups of samples. |
| Randomization | N/A (no experimental groups) |
| Blinding | N/A (no experimental groups) |

# Reporting for specific materials, systems and methods

We require information from authors about some types of materials, experimental systems and methods used in many studies. Here, indicate whether each material, system or method listed is relevant to your study. If you are not sure if a list item applies to your research, read the appropriate section before selecting a response.

## Materials & experimental systems

| n/a | Involved in the study |
|---|---|
| ☒ | ☐ Antibodies |
| ☐ | ☒ Eukaryotic cell lines |
| ☒ | ☐ Palaeontology and archaeology |
| ☒ | ☐ Animals and other organisms |
| ☒ | ☐ Human research participants |
| ☒ | ☐ Clinical data |
| ☒ | ☐ Dual use research of concern |

## Methods

| n/a | Involved in the study |
|---|---|
| ☒ | ☐ ChIP-seq |
| ☒ | ☐ Flow cytometry |
| ☒ | ☐ MRI-based neuroimaging |

## Eukaryotic cell lines

Policy information about cell lines

| | |
|---|---|
| Cell line source(s) | The mES cells used for this study were the 46C line, a Sox1–GFP derivative of E14tg2a and gift from D. Henrique. |
| Authentication | mES cell dentity was confirmed at the time of cryoblock creation by morphology and by confirming GFP expression after neural differentiation. |
| Mycoplasma contamination | mES cells were routinely tested for mycoplasma contamination and were found to be negative. |
| Commonly misidentified lines (See ICLAC register) | Neither 46C mES cells nor the parental line (E14tg2a) are listed on the ICLAC register |

