## [Peer Review File · Nature Methods]

Peer Review Information

Manuscript Title: Multiplex-GAM: genome-wide identification of chromatin contacts yields insights not captured by Hi-C

Corresponding author name(s): Ana Pombo

Editorial Notes:

Reviewer Comments & Decisions:

Decision Letter, initial version:

Dear Dr Pombo,

Your Article entitled "Multiplex-GAM: genome-wide identification of chromatin contacts yields insights not captured by Hi-C." has now been seen by 3 reviewers, whose comments are attached. While they find your work of potential interest, they have raised serious concerns which in our view are sufficiently important that they preclude publication of the work in Nature Methods, at least in its present form.

As you will see, the reviewers raise serious concerns about general applicability of multiplex-GAM and its comparison against single cell Hi-C.

Should further experimental data allow you to fully address these criticisms we would be willing to look at a revised manuscript (unless, of course, something similar has by then been accepted at Nature Methods or appeared elsewhere). This includes submission or publication of a portion of this work somewhere else. We hope you understand that until we have read the revised paper in its entirety we cannot promise that it will be sent back for peer-review.

If you are interested in revising this manuscript for submission to Nature Methods in the future, please contact me to discuss your appeal before making any revisions. Otherwise, we hope that you find the reviewers' comments helpful when preparing your paper for submission elsewhere.

Sincerely,
Lei

Lei Tang, Ph.D.
Senior Editor
Nature Methods

Reviewers' Comments:

Reviewer #1:

Remarks to the Author:

I apologise to the authors if I have misunderstood their method, but it is not clear to me why they think their method offers advantages in comparison to the existing Hi-C experiments carried out on either a large population of cells or single cells. I therefore think it would be helpful if they could set out the pros and cons of the different methods more clearly and also include a more detailed comparison with single cell Hi-C so that we can understand where they see the benefits.

As I understand their method, they take a slice through a single cell and then sequence all the DNA from within that single slice. Thus, they do not directly determine contacts between different DNA sequences (as in Hi-C), but instead identify all the DNA sequences in that slice of the cell. For each individual DNA sequence detected, a few of the other DNA sequences they detect will be interacting with it, but most will not: as far as I can tell they could be as far away as on the opposite sides of the cell. To try and identify those sequences that do interact, they collect data from many different slices through different single cells to work out which sequences tend to be found together more often.

I can see that the GAM method has advantages of not needing to detect physically ligated sequences as in conventional Hi-C experiments, and that using this method one can identify features such as TADs and compartments that are also identified in Hi-C experiments. However, it appears to me to be a very inefficient way of identifying TADs and compartments because the real interacting sequences are detected together with a lot of sequences that don't interact (i.e. noise). This then needs to be filtered out. In contrast, in conventional Hi-C experiments carried out on a large number of cells one directly determines contacts – a contact probability map with high signal to noise is therefore much more easily obtained. The authors also don't compare their method with more recent experiments such as Micro-C, which provide higher resolution than Hi-C.

However, GAM is inherently a single cell method, and I would have thought the more logical comparison would be with single cell Hi-C. Moreover, single cell Hi-C experiments appear to combine some of the advantages of conventional Hi-C and GAM. In Hi-C experiments carried out on single cells all the data collected is relevant to that particular cell and the way the DNA folds in that particular cell. This is helpful because single cell Hi-C experiments by a number of groups have shown that the folding of

chromosomes and how they interact with each other is completely different in every cell. This means that it is not possible to use data from conventional Hi-C experiments collected on a large number of cells to determine how the DNA folds – because the data you collect is an average from many different cells each with a very different structure, and you cannot work out which contacts come from which cell. (The GAM method may have a similar limitation because as implemented it appears that you need to combine slices from many different cells.)

Because you can use single cell Hi-C data to calculate a three-dimensional structure you not only identify the regions that contact each other resulting in a chemical ligation, but you also identify all the other sequences that are close by in the three-dimensional folded structure, but which do not result in chemical ligations in the experiment. Thus, in a 3D structure calculated from single cell Hi-C data you determine many multiway contacts directly. In principle, the GAM method also identifies DNA sequences that are close to each other, but which do not necessarily make a chemical contact. However, the disadvantage is that one also collects a lot of noise corresponding to DNA sequences that could be on the other side of the cell. If you look at good single cell Hi-C datasets deposited in the last few years it is possible to work out how close/far apart different sequences are within the structure, even if they are on opposite sides of the cell when there is no possibility of a chemical ligation occurring.

In summary, it is just not clear to me why the authors think the GAM method offers advantages over a combination of Hi-C experiments carried out on a population of cells and single cells. I can see that one can use vitrification and avoid the need for fixation, but some groups have shown that one can do Hi-C experiments without fixation. The authors suggest that GAM offers advantages for picking up single cells compared to Hi-C experiments. This could be true, but on the other hand, there is a lot of work nowadays to efficiently carry out single cell experiments such as RNA-seq and ATAC-seq in cells from tissues, which make me feel that the technology for isolating single cells for single cell Hi-C experiments should not be that difficult. Moreover, some groups have developed highly automated single cell Hi-C protocols that have been implemented on liquid handling robots such that it is now possible to very readily generate thousands of single cell Hi-C datasets. Thus, one can determine many 3D structures for a particular cell type and sample the population of structures formed. As I say, I may have misunderstood the GAM method, but it is not immediately obvious to me why it is advantageous. There are now single cell Hi-C datasets from mES cells deposited for large numbers of cells and I would like to see more of a comparison of GAM with single cell Hi-C, and more of a discussion by the authors regarding the pros and cons and how they relate to the results they observe so that we can understand where the advantages lie.

Reviewer #2:

Remarks to the Author:

In this manuscript, Beagrie and Thieme et al. present Multiplex Genome Architecture Mapping (multiplex-GAM), an evolution of their Genome Architecture Mapping method published in 2017. In multiplex-GAM, the DNA from multiple microdissected nuclear slices are simultaneously sequenced in a single PCR tube. The authors also present work on their mathematical model for GAM/multiplex-GAM data, SLICE, adding parameters to model chromatin interaction information from multiple microdissected nuclear slices per tube. After combining GAM and multiplex-GAM data sets, Beagrie and Thieme et al. proceed to compare GAM/multiplex-GAM data to data generated via Hi-C, detecting similar TADs, compartments, and “strong” contacts in data sets from the two methods. The authors demonstrate that chromatin interactions given more weight by GAM connect genomic loci bound by CTCF, transcription factors, phosphorylated RNAPII, and enhancers/super enhancers. In contrast, chromatin interactions given more weight by Hi-C are somewhat enriched in heterochromatic histone modifications. The authors find that, in comparison to GAM/multiplex-GAM data, ligation frequencies measured by Hi-C are lower between genomic loci that form complex multiway interactions. These genomic loci often comprise super enhancers and other forms of chromatin interaction within A compartments, indicating that multiway interactions are underrepresented in Hi-C data. While previous work asserted an underrepresentation of multiway interactions, the current study is the first to provide empirical evidence for such an effect.

Comments:

There are no analyses (or mention) of inter-chromosomal chromatin interactions in the study. How does GAM/multiplex-GAM compare to Hi-C in this regard? It would be useful to the community to know the answer to that question. For example, the authors could display panels of SLICE predictions for the numbers of NPs needed to achieve inter-chromosomal interaction maps of a given resolution. The authors could also examine how inter-chromosomal interaction maps compare between multiplex-GAM and Hi-C; for example, given the apparent underrepresentation of multiway A-compartment contacts in Hi-C data, would we expect to see a higher number of A-compartment trans interactions in GAM/multiplex-GAM data?

The choice of window size is critical, since it potentially impacts all of the downstream analysis. I am not very convinced by the "erosion analysis" (Supp Fig 3C) that supports the use of 40kb windows for the analysis. In particular, comparison to a previous GAM dataset is not obviously the right way to select the resolution. I recognize that choosing the resolution/bin size is non-trivial, but given how important it is, it seems like the manuscript should spend more effort justifying this particular choice.

A major missing piece of the GAM analysis toolkit is a way to identify a collection of significant contacts, subject to FDR control. Established methods, such as fit-hi-c or HiCDC+, have been developed for Hi-C data. To get around this problem, the authors replace these established methods with a reasonable but fairly ad hoc z-score-based normalization scheme, which can be applied to both data types. On the face

of it, this scheme seems reasonable, but it does not take into account some of the underlying statistical properties of Hi-C data. I would like to be convinced that the z-score-based scheme developed here does not lead to some of the observed biases. To do this, I suggest repeating the comparative analyses using z-scores from HiCDC+. If the overall results are consistent with what is reported in the initial draft of the manuscript, then this confirmatory analysis can be relegated to the supplement. But if the results are different, then some investigation of the source of the differences is warranted.

54-58 (main)

"Although single-cell variants of Hi-C are available, they require purified, disaggregated cell suspensions which can be unachievable for rare cell types embedded within complex tissues."

I don't understand the point here. One benefit of single cell analysis in general is that rare cell types can be analyzed by sequencing all cells in a complex tissue and then "isolating" the rare cell type in silico.

113 / 126 (main)

Please be consistent about how m^* is described -- number of GAM samples or number of tubes. I personally think the latter is clearer terminology.

152–157 (main)

"As our current GAM pipeline does not distinguish these 'cytoplasmic profiles' from NPs, we calculated that if we dissected four cresyl violet stained profiles into each sequencing tube, on average three of those four profiles would intersect the nucleus (Supplementary Fig. 2d)."

Here, a panel in which the authors show quantification of their scoring may be helpful for readers.

189 (main)

How do you know that each contains three NPs on average? Does this involve extrapolating based on the simulation showing that 23% of slices contain no NP?

190–191 (main)

Regarding Fig. S3a, are the varying distributions of mapped reads amid experiments #1–11 really "comparable"?

199 (main)

Before erosion, the Pearson correlation is saturated at slightly above 0.6. While the normalized linkage disequilibrium matrices in Fig. 2a and Fig. 2b appear somewhat similar to each other, is it to be expected that the correlation values would be higher?

202 (main)

Are the correlation coefficient values indicative of the combined 1NP+3NP libraries "[retaining] the quality of 3NP libraries?"

211 (main)

The notion of a "prominent interaction" is not properly defined.

250 (main)

In Fig. S5e, there are data for repetitive elements such as tRNAs and LINEs, but there do not seem to be data for SINEs. Did the authors intend to mention "LINEs" and not "SINEs" here?

251–252 (main)

"In most cases, features found at common boundaries are also found at boundaries identified only in GAM or only in Hi-C except with lower enrichment..."

The authors can aid readers by referring to explicit examples in the figure or supplementary figure. For example, we seem to see this with the phospho-RNAPII data sets and perhaps with the K79me2 data set.

256-259 (main)

This sentence is difficult for me to make sense of. GAM detects more contacts between TADs than what -- Hi-C? I don't understand the logic of the "consistent with" phrase: how does the first half of the sentence imply the second half?

295-296 (main)

The English here is imprecise. Does this mean that you identified all sites with delta z-score < 1 and then subsequently identified the ones that are strongest? What is the criterion for strongest? Mean z-score?

322 (main)

The terminology "differentially detected" is somewhat misleading, since the threshold for "detection" has been arbitrarily set. To complement the analysis in Figure 5a, I think it's worth characterizing the distributions of z-scores (Hi-C and GAM) associated with each feature pair, irrespective of the arbitrary threshold.

348-352 (main)

This is a provocative finding, and one that's deserving of elaboration (particularly the assertion that CTCF contacts occurring with enhancers and active genes are underrepresented in Hi-C data). Also, following on the complexity analyses, do these data indicate that, given their lower complexity, heterochromatin condensates are more homogenous than A-compartment condensates associated with CTCF, gene transcription, enhancers, etc.? Were the authors to follow up on this, it would be very useful information for the field.

612 (main)

Why are we still using mm9? It was superseded by mm10 in Dec 2011.

47–48 (supplementary)

"Left and middle bars are the same as in Supplementary Fig. 1e." There does not appear to be a panel e in Fig. S1.

In Figure 3b/c, why do the middles of each Venn diagram contain two different numbers with a slash between? Also, I think this data would be presented better as a colored bar.

<https://medium.com/@raynamharris/bar-plots-as-venn-diagram-alternatives-d25888369c84>

Reviewer #3:

Remarks to the Author:

In this paper, Pombo and colleagues present an improved and more highly multiplexed version of GAM. Because the resolution of a GAM dataset is dependent on the number of slices measured, this also enables generation of higher resolution datasets that was previously possible. The authors show that these datasets are able to uncover key features of genome structure that are not captured by proximity-ligation methods.

Overall, I think this manuscript is a great candidate for publication in Nature Methods. GAM represents one of the truly breakthrough technologies in the 3D genome architecture field and its initial adoption (following the authors' 2017 paper) led to critical new insights into multi-way genome contacts. Despite the clear importance of this method and the data it generates, the main limitation of this approach that has restricted its widespread adoption are the technical challenges associated with generating a GAM dataset. Specifically, it requires generating hundreds or thousands of nuclear slices, collecting each within an individual well and performing whole genome amplification and other molecular manipulations. In the current manuscript, the authors present an adaption of the GAM method that dramatically simplifies the experimental procedure by enabling multiplexed handling and measurements of multiple nuclear slices. This will clearly enable more widespread adoption of this method for many studies.

My only real critique of the method is that the multiplexing procedure relies on the low probability of inter-chromosomal contacts to enable demultiplexing of nuclear slices. While this may work as a general feature (as the authors clearly show), it would fail to map inter-chromosomal contacts. Even though such interactions are rare, they do still occur and are known to be critical for genome organization. I don't think this is a deal breaker given the general capabilities of this method, but the authors should note this limitation more explicitly.

One additional minor comment/suggestion. I think the manuscript would have much more impact as a significantly shorter manuscript.

Author Rebuttal to Initial comments

Please find below our point-by-point response to the reviewers' requests.

Reviewer #1:

I apologise to the authors if I have misunderstood their method, but it is not clear to me why they think their method offers advantages in comparison to the existing Hi-C experiments carried out on either a large population of cells or single cells. I therefore think it would be helpful if they could set out the pros and cons of the different methods more clearly and also include a more detailed comparison with single cell Hi-C so that we can understand where they see the benefits.

1. *We thank the reviewer for encouraging us to state the pros and cons of GAM more clearly relative to bulk and scHi-C. We have included a new summary table to address this point (Supplementary Table 1), and have mined scHi-C data and the models derived from it to assess TAD border identification (Supplementary Fig. 7) and multiway contacts (Fig. 5).*

I can see that the GAM method has advantages of not needing to detect physically ligated sequences as in conventional Hi-C experiments, and that using this method one can identify features such as TADs and compartments that are also identified in Hi-C experiments. However, it appears to me to be a very inefficient way of identifying TADs and compartments because the real interacting sequences are detected together with a lot of sequences that don't interact (i.e. noise). This then needs to be filtered out. In contrast, in conventional Hi-C experiments carried out on a large number of cells one directly determines contacts – a contact probability map with high signal to noise is therefore much more easily obtained.

2. *As the reviewer notes, GAM captures information about all the DNA present in a single plane through the nucleus. This information provides a snapshot of genome architecture and has its own unique advantages, for example allowing the method to directly measure aspects of chromatin folding not captured by bulk Hi-C such as radial position and compaction (Beagrie et al. 2017). More recently GAM detected decondensation (melting) of long neuronal genes when most active (Winick-Ng et al. 2021) which is also not captured by Hi-C (see page 59 of peer-review document).*

We have demonstrated that contact signal in experimental GAM data is well above noise from short to very long range, capturing compartments, TADs, pairwise and multi-way contacts (Beagrie et al. 2017, Winick-Ng et al. 2021). In addition, to formally compare

GAM and Hi-C, we have recently published an in-depth theoretical exploration of these aspects in Nature Methods (Fiorillo et al. 2021) which demonstrated the better performance of GAM compared with Hi-C or SPRITE. After generating simulated Hi-C and GAM datasets from the same set of 3D polymer models, we found that GAM more accurately captures the average distance between DNA loci and has a better signal-to-noise ratio than Hi-C across distances, but particularly at large genomic distances (>1Mb) where GAM performs nearly an order of magnitude better. As this work is now published, we have referenced it appropriately in the revised manuscript.

“real interacting sequences are detected together with a lot of sequences that don’t interact (i.e. noise)”.

The reviewer seems concerned that GAM may be less able to identify biologically important DNA contacts than Hi-C due to the presence of non-contacting DNA within the same slice. This can be a common misunderstanding of how contacts are extracted from GAM data. Contrary to the reviewer’s statement, the fraction of nuclear slices that contains any two loci (as measured by GAM) is a simple function of their 3D distance. Genomic regions that are physically distant are only rarely found in the same thin nuclear slices. The SLICE model (originally presented in Beagrie et al. 2017 and updated in this manuscript) makes use of this statistical property of GAM data to identify loci interacting above background frequencies. In contrast, Hi-C experiments capture at most two interacting partners for any given DNA fragment, chosen from the pool of nearby DNA sequences in a stochastic manner, and it is not true that the capture is restricted only to contacts, especially not in in-situ ligation methods where many captured ligation events will be random. Capture by ligation is much more difficult to model mathematically than GAM-type sampling, and therefore distinguishing between interesting contacts and background noise is more challenging in ligation-based methods. There is also no direct way to assess from Hi-C data the physical proximity of loci that contact at larger distances (e.g. around a nuclear speckle or nucleoli). In contrast, GAM directly senses physical distance from the likelihood of finding any two genomic regions in the same collection of slices.

The authors also don’t compare their method with more recent experiments such as Micro-C, which provide higher resolution than Hi-C.

Our present manuscript does not aim to assess the performance of GAM technology across all genomic length scales and resolutions. We note that the biases demonstrated in 3C-based methods to preferential capture simple, over multiway, contacts, due to ligation dependency, can be extrapolated to affect all ligation-dependent methods at any genomic length scale, including micro-C. As with C-based methods, high resolution maps will become possible within GAM technologies with further ongoing refinements.

However, GAM is inherently a single cell method, and I would have thought the more logical comparison would be with single cell Hi-C. Moreover, single cell Hi-C experiments appear to combine some of the advantages of conventional Hi-C and GAM. In Hi-C experiments carried out on single cells all the data collected is relevant to that particular cell and the way the DNA folds in that particular cell. This is helpful because single cell Hi-C experiments by a number of groups have shown that the folding of chromosomes and how they interact with each other is completely different in every cell. This means that it is not possible to use data from conventional Hi-C experiments collected on a large number of cells to determine how the DNA folds – because the data you collect is an average from many different cells each with a very different structure, and you cannot work out which contacts come from which cell. (The GAM method may have a similar limitation because as implemented it appears that you need to combine slices from many different cells.)

- 3. We are glad the reviewer has appreciated that GAM retains single-cell information in a way that bulk-Hi-C does not. The fact that the same protocol is useful for bulk or single-cell analyses is a key advantage of GAM. However, we do not share the reviewer's appraisal that bulk data are not useful, nor their assessment of the extent of cell-to-cell variability. Many major principles of chromatin folding have been identified from bulk datasets (TADs, compartments, CTCF loops) which would not be possible if DNA folding was completely different in each cell. Single-cell data is most useful when there are enough cells for commonalities to be drawn out, for example others have recently shown at least 300 cells are needed to reliably recover chromatin loops from single-cell Hi-C datasets (Yu et al. Nature Methods 2021; PMID 34446921).*

Because you can use single cell Hi-C data to calculate a three-dimensional structure you not only identify the regions that contact each other resulting in a chemical ligation, but you also identify all the other sequences that are close by in the three-dimensional folded structure, but which do not result in chemical ligations in the experiment. Thus, in a 3D structure calculated from single cell Hi-C data you determine many multiway contacts directly.

- 4. In this manuscript, we show that ligation events within multi-way interactions are consistently lower when measured by Hi-C, whilst these same regions show strong interactions in GAM. This finding was theoretically predicted by earlier work (O'Sullivan et al 2013). Several groups have reported generating 3D structures from single-cell Hi-C data. The question then is whether 3D modelling of single-cell Hi-C datasets is able to position genomic regions involved in multiway contacts close in 3D space even though direct ligation events between the regions are underestimated. We added an analysis to the revised manuscript in which we assess a published dataset of 3D structures generated from single-cell Hi-C data from individual mES cells. We find that the 3D structures modelled from sparse pairwise contacts captured by scHi-C data contain less than 10%*

of all the three-way contacts observed in GAM data from mES cells and perform no better than bulk Hi-C for predicting three-way contacts (Fig. 5f,g).

In principle, the GAM method also identifies DNA sequences that are close to each other, but which do not necessarily make a chemical contact. However, the disadvantage is that one also collects a lot of noise corresponding to DNA sequences that could be on the other side of the cell. If you look at good single cell Hi-C datasets deposited in the last few years it is possible to work out how close/far apart different sequences are within the structure, even if they are on opposite sides of the cell when there is no possibility of a chemical ligation occurring.

5. *Both GAM and Hi-C measure contacts based on proximity, irrespectively of whether the loci make a chemical contact. In GAM, the contact is measured by how often loci co-segregate in thin slices, in Hi-C by the ligation of free DNA ends, themselves distant from the true contact, in a clump of digested chromatin fragments. To explore this point in more detail, we examined the ability of Hi-C and GAM to capture distances between DNA loci (including loci that do not directly interact) in detail in Fiorillo et al. 2021. We have cited Fiorillo et al. 2021 in the revised manuscript and noted that GAM was found to perform better than Hi-C at capturing average distance matrices ($r_s = -0.89$ for Hi-C; $r_s = -0.99$ for GAM). To strengthen this point, we have repeated the same analyses here for 500 in-silico cells with the typical range of detection efficiencies in scHi-C and GAM datasets. We find that GAM outperforms scHi-C at all detection efficiencies in the correlation of the obtained contact matrix to the ground truth distance matrix (Figure I). It is important to note that GAM datasets typically have >85% capture efficiency (Beagrie et al. 2017, and here), whereas sc-Hi-C typically has 5% up to ~50% in the best cases reported so far (Tan et al. 2018).*

Figure I: a, Average distance matrix of an ensemble of 3D polymer structures of the mESC Sox9 locus in 500 in-silico cells (Fiorillo et al. 2021). b, Heatmaps of Hi-C performed on those 500 in-silico cells with

an efficiency of 0.001, 0.05 and 0.5. Those efficiencies span the range from poor to current best experimental values currently available in the literature. The Spearman correlations between the average distance matrix and the in-silico Hi-C matrices are respectively -0.03, -0.67 and -0.86. **c**, Heatmaps of non-multiplex GAM performed on those 500 in-silico cells with INP per sample and efficiency of 0.1, 0.6 and 0.97, corresponding to a range from poor to best GAM experimental efficiencies (0.85 was obtained in the first GAM dataset, Beagrie et al. 2017). The Spearman correlations between the distance and the in-silico GAM matrices are -0.14, -0.72 and -0.90. **d**, Distribution of Spearman correlations between the distance matrix against replicates of Hi-C over 500 in-silico cells and **e**, of GAM.

In summary, it is just not clear to me why the authors think the GAM method offers advantages over a combination of Hi-C experiments carried out on a population of cells and single cells.

6. *We hope that the revisions included have clarified the advantages of GAM over Hi-C carried out in bulk or single cells. We contend that (a) the under-detection of complex contacts in Hi-C due to ligation biases, which are not effectively circumvented in 3D models inferred from scHi-C data (Fig. 5), and (b) the blindness of Hi-C to many contacts established by regions that are highly enriched for gene and enhancer activity (Fig. 4), are sufficient to argue for an urgent need for additional mapping of 3D genome structure by ligation-free methods, such as GAM.*

I can see that one can use vitrification and avoid the need for fixation, but some groups have shown that one can do Hi-C experiments without fixation.

7. *Our current GAM experiments are run in extremely well preserved samples, which are fixed as for electron microscopy. We have not argued that fixation per-se is a limitation of Hi-C, only that chromatin is by definition disrupted with weak or no fixation used in Hi-C mapping. We have added the reference describing Hi-C experiments conducted without crosslinking agents (Rao et al. 2014; Brant et al. 2016) to the Discussion section of the manuscript.*

The authors suggest that GAM offers advantages for picking up single cells compared to Hi-C experiments. This could be true, but on the other hand, there is a lot of work nowadays to efficiently carry out single cell experiments such as RNA-seq and ATAC-seq in cells from tissues, which make me feel that the technology for isolating single cells for single cell Hi-C experiments should not be that difficult.

8. *We thank the reviewer for the opportunity to clarify this point in the revised manuscript, as we believe this aspect to be one of the most important advantages of GAM over Hi-C and there are multiple issues to address. The reviewer is correct that most metazoan*

tissues can be dissociated into single cells by some combination of enzymatic digestion and mechanical agitation. However, this process can alter cell physiology (Van den Brink et al. 2017 PMID 28960196) and compromise cell identification (Millard et al. 2021 PMID 34818538). Within the same tissue, there may be a mix of cell types that can only be preserved by very gentle extraction protocols and others that require much more robust methods (Denisenko et al. 2020 PMID 32487174). These technical issues can present a serious impediment when studying diseased tissue, where cells may be even more fragile (Korin et al. 2021 PMID 34282333). Entirely eliminating the need to disturb cellular structures by mechanical or enzymatic dissociation is therefore a substantial advantage on its own.

Once a disaggregated tissue is obtained, one has two options. The first is to proceed with scHi-C library generation for all obtained cells and extract data for the cell types of interest by clustering related cells after sequencing. This approach is viable where the cell type of interest is a major component of the targeted tissue, but quickly becomes prohibitively expensive for rare cell types of special interest. The expense can also be compounded when the cell type of interest is sensitive to disaggregation. As an example, we recently used GAM to profile chromatin folding in dopaminergic neurons from intact mouse brains (Winick-Ng et al. Nature 2021). Whilst these cells make up 15% of the midbrain, only 1-2% of recovered cells in single-nucleus RNA-seq and ATAC-seq experiments are DNs (LaManno et al. 2016 PMID: 27716510; Tuesta et al. 2019 PMID: 31175277). To generate single-cell Hi-C data from 200 DNs would therefore require the processing, sequencing and analysis of 20,000 midbrain cells with sequencing costs alone 10-20x higher than those required to generate the published GAM dataset mapping 2500 DNs.

The second option is to enrich for the cell type of interest after tissue dissociation by flow sorting. This requires marker genes that are expressed on the cell surface, which are not available for every cell type. Not all cell types are robust enough to survive extrusion through the high-pressure nozzle, and cell sorting is also not well suited to large or irregularly shaped cells like neurons.

Most importantly, GAM preserves information about the geographical origins of each cell within the original tissue. Nature Methods named spatially-resolved transcriptomics as its “Method of the Year 2020” because of the emerging understanding that spatial information is crucial for proper understanding of cell identity and function. Spatial differences in gene expression are increasingly being found within cell types previously reported to be homogenous by single-cell studies performed from dissociated tissues (e.g. Rodrigues et al. 2019). The same phenomenon has recently been observed in spatially-resolved ATAC-seq data (Deng et al. 2021; doi:10.1101/2021.06.06.447244) and FISH experiments have shown that enhancer-promoter contacts can also be specific to spatial

regions of one tissue (Williamson et al. 2016).

In short, we thank the reviewer for drawing attention to the fact that many substantial advantages of GAM over single-cell Hi-C were not mentioned in our original manuscript. We have addressed these aspects of the method in a new summary table (Supplementary Table 1, see point 1).

Moreover, some groups have developed highly automated single cell Hi-C protocols that have been implemented on liquid handling robots such that it is now possible to very readily generate thousands of single cell Hi-C datasets. Thus, one can determine many 3D structures for a particular cell type and sample the population of structures formed. As I say, I may have misunderstood the GAM method, but it is not immediately obvious to me why it is advantageous. There are now single cell Hi-C datasets from mES cells deposited for large numbers of cells and I would like to see more of a comparison of GAM with single cell Hi-C, and more of a discussion by the authors regarding the pros and cons and how they relate to the results they observe so that we can understand where the advantages lie.

9. *We thank the reviewer for pointing out that, for researchers interested in the study of rare cell types, comparisons between GAM and single-cell Hi-C are likely to be very useful. We have addressed this point via multiple approaches:*
 - a. *We now cite our theoretical comparison between GAM and single-cell Hi-C (Fiorillo et al. 2021)*
 - b. *We cite recent study showing that 400 cells are required for robust detection of long-range loops in scHi-C datasets (Yu et al. 2021; PMID 34446921).*
 - c. *We have added an analysis of TAD boundaries identified by scHi-C (Supplementary Fig. 7b,d). We find that 100-200 cells are required for robust detection of TAD boundaries, therefore the amount of input material required for scHi-C and for GAM is comparable.*
 - d. *We have added an analysis of multi-way contacts in scHi-C. We find that scHi-C performs no better than bulk Hi-C at detecting such interactions (Fig. 5f,g).*
 - e. *We have included scHi-C in our new comparison table of methods for measuring chromatin folding (Supplementary Table 1).*

Reviewer #2:

Remarks to the Author:

In this manuscript, Beagrie and Thieme et al. present Multiplex Genome Architecture Mapping (multiplex-GAM), an evolution of their Genome Architecture Mapping method published in 2017. In multiplex-GAM, the DNA from multiple microdissected nuclear slices are simultaneously sequenced in a single PCR tube. The authors also present work on their mathematical model for GAM/multiplex-GAM data, SLICE, adding parameters to model chromatin interaction information from multiple microdissected nuclear slices per tube. After

combining GAM and multiplex-GAM data sets, Beagrie and Thieme et al. proceed to compare GAM/multiplex-GAM data to data generated via Hi-C, detecting similar TADs, compartments, and “strong” contacts in data sets from the two methods. The authors demonstrate that chromatin interactions given more weight by GAM connect genomic loci bound by CTCF, transcription factors, phosphorylated RNAPII, and enhancers/super enhancers. In contrast, chromatin interactions given more weight by Hi-C are somewhat enriched in heterochromatic histone modifications. The authors find that, in comparison to GAM/multiplex-GAM data, ligation frequencies measured by Hi-C are lower between genomic loci that form complex multiway interactions. These genomic loci often comprise super enhancers and other forms of chromatin interaction within A compartments, indicating that multiway interactions are underrepresented in Hi-C data. While previous work asserted an underrepresentation of multiway interactions, the current study is the first to provide empirical evidence for such an effect.

10. We are happy the reviewer appreciated our analyses providing the first evidence for underrepresentation of multi-way interactions in Hi-C data since such an effect was only predicted theoretically (O’Sullivan et al 2013) but not previously demonstrated in practice. We were also pleased that the reviewer fully appreciated the differences we find in detection power between GAM and Hi-C.

Comments:

There are no analyses (or mention) of inter-chromosomal chromatin interactions in the study. How does GAM/multiplex-GAM compare to Hi-C in this regard? It would be useful to the community to know the answer to that question. For example, the authors could display panels of SLICE predictions for the numbers of NPs needed to achieve inter-chromosomal interaction maps of a given resolution. The authors could also examine how inter-chromosomal interaction maps compare between multiplex-GAM and Hi-C; for example, given the apparent underrepresentation of multiway A-compartment contacts in Hi-C data, would we expect to see a higher number of A-compartment trans interactions in GAM/multiplex-GAM data?

11. We agree with the reviewer that inter-chromosomal interactions are an understudied aspect of nuclear biology that deserve further attention. However, it is difficult to compare the relative merits of different methods for their detection when the field is yet to reach a consensus on the most important aspects, and when high numbers of ligation events between chromosomes are usually considered to indicate poor dataset quality.

For example, multiple groups have reported that specific interactions between different chromosomes are extremely rare, and their biological relevance is unclear. There is more evidence that particular loci prefer to reside within, on the surface of or fully outside their respective chromosome territory, and there are clear and testable hypotheses about how this affects access to other nuclear structures. Other groups have

had success accessing this information from scHi-C datasets through 3D modelling (Tan et al. 2018), but extracting such properties from bulk Hi-C data has proved to be extremely challenging. We have had some success assessing these propensities in neuronal GAM datasets (Winick-Ng et al. Nature 2021). It is known that mES cells have particularly weak inter-chromosomal contacts (Tan et al. 2018). We therefore plan to examine trans-contacts using matched Hi-C and GAM datasets from a cell type with more stable trans-contacts in a dedicated manuscript.

The choice of window size is critical, since it potentially impacts all of the downstream analysis. I am not very convinced by the "erosion analysis" (Supp Fig 3C) that supports the use of 40kb windows for the analysis. In particular, comparison to a previous GAM dataset is not obviously the right way to select the resolution. I recognize that choosing the resolution/bin size is non-trivial, but given how important it is, it seems like the manuscript should spend more effort justifying this particular choice.

- 12. We thank the reviewer for drawing our attention to this important aspect of experimental design, which will be of relevance to other groups looking to establish GAM in their laboratories. We use our statistical model SLICE to set an appropriate resolution for each GAM dataset, the most important aspect of which is to establish the resolution at which our detection of true-positive windows drops below 80%.*

The reviewer's comment helped us realise that a simple heuristic measure for the minimum resolution of a GAM dataset would be very useful for the community. Many Hi-C studies establish the minimal resolution of a dataset by calculating the resolution at which >80% of bins contain at least 1000 contacts. Along these lines, we now advise that the minimum resolution for a GAM dataset is one in which >80% of bins appear in at least 25 slices. We have incorporated this guidance into the revised manuscript.

We have also developed a new tool for our "GAMtools" suite, which uses both SLICE analysis and our new measure to advise the user on a suitable minimum resolution for any particular GAM dataset. We are working on a manuscript dedicated to this software suite (previous revision available at bioRxiv) which we believe would be the appropriate place to explore these measures in greater depth.

A major missing piece of the GAM analysis toolkit is a way to identify a collection of significant contacts, subject to FDR control. Established methods, such as fit-hi-c or HiCDC+, have been developed for Hi-C data.

- 13. We fully agree with the reviewer that statistical significance is a critical point to derive "true contacts". We developed the SLICE model to fulfil exactly this analytical need. As described in our 2017 paper, the original SLICE model detected significant contacts but*

did not include mechanisms for FDR control. We now add FDR control to the updated version of SLICE described in this manuscript. We tested whether this more stringent set of significant contacts was enriched for similar biological features, such as active or enhancer regions, and found it to be the case (Supplementary Fig. 6c in the revised manuscript). We thank the reviewer for drawing our attention to the ambiguity in the original manuscript and have clarified these points in the revised text.

To get around this problem, the authors replace these established methods with a reasonable but fairly ad hoc z-score-based normalization scheme, which can be applied to both data types. On the face of it, this scheme seems reasonable, but it does not take into account some of the underlying statistical properties of Hi-C data. I would like to be convinced that the z-score-based scheme developed here does not lead to some of the observed biases. To do this, I suggest repeating the comparative analyses using z-scores from HiCDC+. If the overall results are consistent with what is reported in the initial draft of the manuscript, then this confirmatory analysis can be relegated to the supplement. But if the results are different, then some investigation of the source of the differences is warranted.

14. We thank the reviewer for suggesting that we examine statistically significant Hi-C contacts. In the revised manuscript, we made use of the significant interactions from Hi-C called by fit-Hi-C and investigate whether they fall in method-specific or common categories. We find that many of the fit-Hi-C interactions fall into the strong-and-common contacts detected by both GAM and Hi-C, or Hi-C-specific categories (Figure 2; Supplementary Figure 8d), which is possibly unexpected as GAM-specific contacts are enriched for complex contacts that are under-detected as pair-wise contacts in Hi-C data.

Following the advice of the reviewer, we further compare observed over expected and rank transformation methods to z-score transformation and show that 300,000 out of 500,000 Hi-C-specific and GAM-specific contacts are captured by all three transformations (Supplementary Fig. 8b).

“Although single-cell variants of Hi-C are available, they require purified, disaggregated cell suspensions which can be unachievable for rare cell types embedded within complex tissues.” I don’t understand the point here. One benefit of single cell analysis in general is that rare cell types can be analyzed by sequencing all cells in a complex tissue and then “isolating” the rare cell type in silico.

Please see point 8 above in response to a similar comment from Reviewer #1.

Please be consistent about how m^* is described -- number of GAM samples or number of tubes. I personally think the latter is clearer terminology.

15. We thank the reviewer for making us aware of this inconsistency. We have now ensured that m^ is described as the minimal number of tubes throughout the revised manuscript.*

"As our current GAM pipeline does not distinguish these 'cytoplasmic profiles' from NPs, we calculated that if we dissected four cresyl violet stained profiles into each sequencing tube, on average three of those four profiles would intersect the nucleus (Supplementary Fig. 2d)." Here, a panel in which the authors show quantification of their scoring may be helpful for readers. How do you know that each contains three NPs on average? Does this involve extrapolating based on the simulation showing that 23% of slices contain no NP?

16. We directly measure by imaging that 23% of slices contain no NP (Supplementary Fig. 4e). We then show that a theoretical bimodal distribution based on this empirical result yields a mean of three NPs per tube (Supplementary Fig. 4f). Finally, we show that the mean number of positive windows detected in each tube matches the predicted value for three NPs per tube (Supplementary Fig. 5b).

Regarding Fig. S3a, are the varying distributions of mapped reads amid experiments #1–11 really "comparable"?

17. We apologise for the confusion. We have corrected the text of the revised manuscript, as we meant to draw attention to the percentage of positive windows rather than the percentage of mapped reads. The percentage of mapped reads is consistent for all experiments except for experiment 10, which we retain in the final dataset because it did not appear to differ from other batches in terms of the percentage of positive windows (Supplementary Fig. 5 in the revised manuscript). This is expected to happen since positive windows are defined by a sufficient read coverage above a data inherent noise level.

Before erosion, the Pearson correlation is saturated at slightly above 0.6. While the normalized linkage disequilibrium matrices in Fig. 2a and Fig. 2b appear somewhat similar to each other, is it to be expected that the correlation values would be higher?

18. We did not have any particular expectation for the Pearson correlation, although it is perhaps worth noting that the two datasets were collected several years apart, from different biological sources and in different physical locations (due to a laboratory move from UK to Germany), all of which could have contributed to the slightly lower correlation coefficient.

Are the correlation coefficient values indicative of the combined 1NP+3NP libraries "[retaining] the quality of 3NP libraries?"

19. *We are aware that the correlation between two halves of a dataset is a rather blunt tool for assessing quality, but we believe it is a useful measure for detecting any additional noise introduced by mixing datasets. By this logic, we show that 3NP samples are slightly more poorly correlated than INP samples (Supplementary Fig. 6a). INP+3NP samples are intermediate between the two (rather than being similar to 3NP samples). We have now clarified this point in the revised text.*

The notion of a "prominent interaction" is not properly defined.

20. *We thank the reviewer for drawing our attention to this point. Prominent interactions (as described in our 2017 paper) are significant interactions identified by SLICE prior to FDR control. For clarity, in the revised manuscript we now refer only to "interactions identified by SLICE".*

In Fig. S5e, there are data for repetitive elements such as tRNAs and LINEs, but there do not seem to be data for SINEs. Did the authors intend to mention "LINEs" and not "SINEs" here?

21. *We thank the reviewer for spotting this error; we had indeed intended to mention "LINEs". This text has been removed from the revised manuscript for concision.*

"In most cases, features found at common boundaries are also found at boundaries identified only in GAM or only in Hi-C except with lower enrichment..."

The authors can aid readers by referring to explicit examples in the figure or supplementary figure. For example, we seem to see this with the phospho-RNAPII data sets and perhaps with the K79me2 data set.

22. *In attempting to make the revised manuscript more concise (see point 31 below) we have moved the figure describing the analysis of TAD boundaries to the supplementary materials and now discuss the results of these analyses in a briefer format.*

This sentence is difficult for me to make sense of. GAM detects more contacts between TADs than what -- Hi-C? I don't understand the logic of the "consistent with" phrase: how does the first half of the sentence imply the second half?

23. *Our aim in this section was merely to point out that the boundaries detected by Hi-C but not by GAM have lower insulation scores, and we have simplified this message in the revised text (page 6).*

The English here is imprecise. Does this mean that you identified all sites with delta z-score < 1 and then subsequently identified the ones that are strongest? What is the criterion for strongest? Mean z-score?

24. We thank the reviewer for pointing out the imprecision at this point. We have clarified in the revised text that we first select sites with delta z-score < 1 and then rank them by the lower of the z-scores from Hi-C and GAM.

The terminology "differentially detected" is somewhat misleading, since the threshold for "detection" has been arbitrarily set. To complement the analysis in Figure 5a, I think it's worth characterizing the distributions of z-scores (Hi-C and GAM) associated with each feature pair, irrespective of the arbitrary threshold.

25. We recognise that the results described rely on comparing the top 5% of contacts by z-score from both GAM and Hi-C. To demonstrate that the observed enrichments are independent from our 5% threshold, we show in the revised manuscript that very similar results are obtained using a less stringent 10% threshold (Supplementary Fig. 9c). Following the reviewer's suggestion, we plot below the z-score distributions from Hi-C and GAM for each feature pair. This data shows that the described differences are not just a characteristic of the extremes but persist over wide ranges of z-score differences (Figure II).

Figure II: Number of all contacts within 4Mb distance with feature pair presence at their anchors. Counts are given for Z-scores observed in GAM or Hi-C in steps of 0.1.

This is a provocative finding, and one that's deserving of elaboration (particularly the assertion that CTCF contacts occurring with enhancers and active genes are underrepresented in Hi-C

data). Also, following on the complexity analyses, do these data indicate that, given their lower complexity, heterochromatin condensates are more homogenous than A-compartment condensates associated with CTCF, gene transcription, enhancers, etc.? Were the authors to follow up on this, it would be very useful information for the field.

26. We appreciate the reviewer's assessment and open mind about our findings. In the revised manuscript, we examine the differential features in further detail and find that the Hi-C specific contacts are enriched for windows that make very few ligation events (Supplementary Fig. 8d). This is somewhat paradoxical. In the case of enhancer+active regions, the underestimation may be due to the possibility that many of these contacts are complex, with each pair-wise contact being lowly represented. In the case of heterochromatin, it may be in addition that restriction enzymes have poor-accessibility to all possible restriction sites. We also suspect that commonly-used matrix normalisation methods over-correct for these poorly-detected regions, an effect which has also been observed by other groups (Chandraross et al. 2020). This normalisation bias may explain the enrichment for H3K9me3/H3K20me3, as these marks are frequently found in regions which do not generate many usable ligation events in Hi-C datasets (see also Supplementary Fig. 8d).

Why are we still using mm9? It was superseded by mm10 in Dec 2011.

27. We agree with the reviewer that the newer Mouse assemblies should be more widely used, however, even in 2021 a plethora of mouse resources is only available in mm9 (e.g.: <http://enhanceratlas.net/scenhancer/>). We are confident that the results we present here will not be dependent on the mouse genome assembly used, but to maximise the utility of the dataset to the wider community, we have added mm10 and mm39 GAM matrices to the GEO record for this manuscript (GSE166381).

In Figure 3b/c, why do the middles of each Venn diagram contain two different numbers with a slash between? Also, I think this data would be presented better as a colored bar.

28. We appreciate the suggestion of using coloured bars to represent this data (now Supplementary Fig. 7b). The insulation score algorithm assigns each boundary a range, therefore one boundary from GAM can rarely overlap two boundaries from Hi-C (or vice versa). This means two slightly different overlap counts can be obtained depending on whether one scores GAM boundaries or Hi-C boundaries with at least one overlap. We now present these two different overlaps separately for clarity.

Reviewer #3:
Remarks to the Author:

In this paper, Pombo and colleagues present an improved and more highly multiplexed version of GAM. Because the resolution of a GAM dataset is dependent on the number of slices measured, this also enables generation of higher resolution datasets that was previously possible. The authors show that these datasets are able to uncover key features of genome structure that are not captured by proximity-ligation methods.

Overall, I think this manuscript is a great candidate for publication in Nature Methods. GAM represents one of the truly breakthrough technologies in the 3D genome architecture field and its initial adoption (following the authors' 2017 paper) led to critical new insights into multi-way genome contacts. Despite the clear importance of this method and the data it generates, the main limitation of this approach that has restricted its widespread adoption are the technical challenges associated with generating a GAM dataset. Specifically, it requires generating hundreds or thousands of nuclear slices, collecting each within an individual well and performing whole genome amplification and other molecular manipulations. In the current manuscript, the authors present an adaptation of the GAM method that dramatically simplifies the experimental procedure by enabling multiplexed handling and measurements of multiple nuclear slices. This will clearly enable more widespread adoption of this method for many studies.

29. We thank the reviewer for their kind words and for their appreciation of the significance of the multiplex-GAM approach, which indeed drastically simplifies the experimental procedure.

My only real critique of the method is that the multiplexing procedure relies on the low probability of inter-chromosomal contacts to enable demultiplexing of nuclear slices. While this may work as a general feature (as the authors clearly show), it would fail to map inter-chromosomal contacts. Even though such interactions are rare, they do still occur and are known to be critical for genome organization. I don't think this is a deal breaker given the general capabilities of this method, but the authors should note this limitation more explicitly.

30. We do not agree with the statement that multiplex-GAM cannot map inter-chromosomal contacts, as this ability depends on the frequency with which any two genomic regions appear within a given physical distance. If two genomic regions from different chromosomes would be consistently found at a given physical distance, more so than any other two other regions from different chromosomes, multiplex-GAM, and the original GAM-INP approach would certainly detect it, given that sufficient numbers of samples were collected. We do not address inter-chromosomal contacts in this manuscript because such contacts are weak in mES cells, and because no matched Hi-C and GAM datasets currently exist for other cell types (see also point 11 above). Further work is necessary to enable a better understanding of inter-chromosomal contacts, their frequency and biological significance.

One additional minor comment/suggestion. I think the manuscript would have much more impact as a significantly shorter manuscript.

31. We agree with the reviewer and have endeavoured to shorten the revised manuscript by moving some figure panels to the supplementary materials and shortening our discussion of the corresponding results in the main text.

Decision Letter, first revision:

Dear Ana,

Thank you for submitting your revised manuscript "Multiplex-GAM: genome-wide identification of chromatin contacts yields insights not captured by Hi-C." (NMEMH-A45365B). It has now been seen by the original referees and their comments are below. The reviewers find that the paper has improved in revision, and therefore we'll be happy in principle to publish it in Nature Methods, pending minor revisions to satisfy the referees' final requests and to comply with our editorial and formatting guidelines.

TRANSPARENT PEER REVIEW

Nature Methods offers a transparent peer review option for new original research manuscripts submitted from 17th February 2021. We encourage increased transparency in peer review by publishing the reviewer comments, author rebuttal letters and editorial decision letters if the authors agree. Such peer review material is made available as a supplementary peer review file. Please state in the cover letter 'I wish to participate in transparent peer review' if you want to opt in, or 'I do not wish to participate in transparent peer review' if you don't. Failure to state your preference will result in delays in accepting your manuscript for publication.

Please note: we allow redactions to authors' rebuttal and reviewer comments in the interest of confidentiality. If you are concerned about the release of confidential data, please let us know specifically what information you would like to have removed. Please note that we cannot incorporate redactions for any other reasons. Reviewer names will be published in the peer review files if the reviewer signed the comments to authors, or if reviewers explicitly agree to release their name. For

more information, please refer to our [FAQ page](https://www.nature.com/documents/nr-transparent-peer-review.pdf).

Thank you again for your interest in Nature Methods Please do not hesitate to contact me if you have any questions.

Best regards,
Lei

Lei Tang, Ph.D.
Senior Editor
Nature Methods

ORCID

Reviewer #2 (Remarks to the Author):

This revision adds a clear and compelling comparison of the Hi-C, scHi-C, and GAM methods; a comparison of scHi-C and multiplex-GAM for the detection of TADs and complex contacts; an analysis of the effect of detectability on the differential detection of genomic regions by Hi-C and multiplex-GAM; an analysis of contacts identified from Hi-C using an independent pipeline (Fit-HiC); and an interesting biological example highlighting the relevance of contacts detected differentially by multiplex-GAM and Hi-C, among other things. Overall, the authors have done a good job of responding to my previous reviews. I particularly like the inclusion of the rule of thumb to select a resolution for GAM, as well as the new tool in GAMtools to suggest an appropriate resolution.

There is, however, one revision that I could not actually find in the manuscript. The authors say "We now add FDR control to the updated version of SLICE described in this manuscript," and they point to Supplementary Figure 6C as an example of this. However, I could not find anywhere in the manuscript or the supplement a description of how the FDR control is carried out. Please add this description and justify whatever assumptions are needed to make it work. It's also not clear to me why FDR control should only be applied to this one supplementary figure. If you know how to control the FDR, why not do it throughout the manuscript?

Regarding rebuttal point #11, It is exciting to consider the prospect of a future study focusing on inter-chromosomal contacts detected with multiplex-GAM. Since two of the three reviewers have brought up the significance of inter-chromosomal contacts and have inquired as to GAM's ability to detect them, as well as the ability to model and analyze them with SLICE, I think it would be good if the authors briefly touched on this topic in the discussion section. Other readers are likely to share the concerns of the reviewers on this point.

Throughout the manuscript, the authors refer to the heterochromatin-associated histone modification "H3K20me3," which I was not familiar with until reading this work. I want to double-check that the authors do not mean heterochromatin-associated histone modification "H4K20me3." A Google search does not turn up much on "H3K20me3," and when it does, it tends to be a typo for "H4K20me3."

Reviewer #3 (Remarks to the Author):

I had minor comments on the initial version of the manuscript and the authors largely addressed my comments. I remain enthusiastic about this paper and the promise that the method holds for the larger 3D structure field. I would be delighted to see it published.

I do want to clarify one of my previous comments that think the authors may have misunderstood about reducing sensitivity for inter-chromosomal contacts. My point was not that multiplexed GAM wouldn't identify them but rather that it would/might have lower sensitivity for them since they tend to be represented by fewer contacts and therefore if the method requires more support to trust observed contacts it will reduce this sensitivity. The authors actually explicitly state this already in the manuscript: "In general multiplex-GAM performs similar to original-109 GAM, but can require increased number of NPs to detect the weakest contacts, or to work at the highest genomic resolutions".

I do think it would be worth explicitly noting this point about inter-chromosomal interactions. In fact, it could just be a parenthetical to that above sentence - just to make this clear to the reader.

Having said this, it is simply a recommendation to the authors, I have no concerns about it being published in its current form.

Author Rebuttal, first revision:

Reviewer #2:

This revision adds a clear and compelling comparison of the Hi-C, scHi-C, and GAM methods; a comparison of scHi-C and multiplex-GAM for the detection of TADs and complex contacts; an analysis of the effect of detectability on the differential detection of genomic regions by Hi-C and multiplex-GAM; an analysis of contacts identified from Hi-C using an independent pipeline (Fit-HiC); and an interesting biological example highlighting the relevance of contacts detected differentially by multiplex-GAM and Hi-C, among other things. Overall, the authors have done a good job of responding to my previous reviews. I particularly like the inclusion of the rule of thumb to select a resolution for GAM, as well as the new tool in GAMtools to suggest an appropriate resolution.

There is, however, one revision that I could not actually find in the manuscript. The authors say "We now add FDR control to the updated version of SLICE described in this manuscript," and they point to Supplementary Figure 6C as an example of this. However, I could not find anywhere in the manuscript or the supplement a description of how the FDR control is carried out. Please add this description and justify whatever assumptions are needed to make it work. It's also not clear to me why FDR control should only be applied to this one supplementary figure. If you know how to control the FDR, why not do it throughout the manuscript?

- 1. We thank the reviewer for spotting this oversight in the previous version of the manuscript and have added the details for FDR control to the methods section. The FDR control is not applied in other sections of the manuscript because it is only applicable to two-way contacts detected by SLICE, not to our differential analysis (Figs 2, 3 & 4) or three-way contacts (Figs 5 & 6).*

Regarding rebuttal point #11, It is exciting to consider the prospect of a future study focusing on inter-chromosomal contacts detected with multiplex-GAM. Since two of the three reviewers have brought up the significance of inter-chromosomal contacts and have inquired as to GAM's ability to detect them, as well as the ability to model and analyze them with SLICE, I think it would be good if the authors briefly touched on this topic in the discussion section. Other readers are likely to share the concerns of the reviewers on this point.

Throughout the manuscript, the authors refer to the heterochromatin-associated histone modification "H3K20me3," which I was not familiar with until reading this work. I want to double-check that the authors do not mean heterochromatin-associated histone modification "H4K20me3." A Google search does not turn up much on "H3K20me3," and when it does, it tends to be a typo for "H4K20me3."

- 2. We again thank the reviewer for picking up on this mistake. We have corrected all occurrences of H3K20me3 to H4K20me3 in the revised manuscript.*

Reviewer #3:

Remarks to the Author:

I had minor comments on the initial version of the manuscript and the authors largely addressed my comments. I remain enthusiastic about this paper and the promise that the method holds for the larger 3D structure field. I would be delighted to see it published.

I do want to clarify one of my previous comments that think the authors may have misunderstood about reducing sensitivity for inter-chromosomal contacts. My point was not that multiplexed GAM wouldn't identify them but rather that it would/might have lower sensitivity for them since they tend to be represented by fewer contacts and therefore if the method requires more support to trust observed contacts it will reduce this sensitivity. The authors actually explicitly state this already in the manuscript: "In general multiplex-GAM performs similar to original-109 GAM, but can require increased number of NPs to detect the weakest contacts, or to work at the highest genomic resolutions".

I do think it would be worth explicitly noting this point about inter-chromosomal interactions. In fact, it could just be a parenthetical to that above sentence - just to make this clear to the reader.

- 3. We thank the Reviewer and appreciate the clarification. We agree that there is a reduction of sensitivity to detect the weakest contacts, which inter-chromosomal in general are. Following this suggestion, we stressed this point in the revised version by adding a parenthesis to that sentence. We fully agree that such reduction does not prevent to detect inter-chromosomal contacts in 3NP GAM datasets since it is possible to find them with the SLICE method. In this regard, we are indeed working on a version of SLICE dedicated to this purpose that will be tested and applied in ongoing projects.*

Having said this, it is simply a recommendation to the authors, I have no concerns about it being published in its current form.

Final Decision Letter:

Dear Ana,

I am pleased to inform you that your Article, "Multiplex-GAM: genome-wide identification of chromatin contacts yields insights not captured by Hi-C", has now been accepted for publication in Nature Methods. Your paper is tentatively scheduled for publication in our July print issue, and will be published online prior to that. The received and accepted dates will be 8 March 2021 and 1 May 2023. This note is intended to let you know what to expect from us over the next month or so, and to let you know where to address any further questions.

Once your paper is typeset, you will receive an email with a link to choose the appropriate publishing options for your paper and our Author Services team will be in touch regarding any additional information that may be required.

Please note that *Nature Methods* is a Transformative Journal (TJ). Authors may publish their research with us through the traditional subscription access route or make their paper immediately open access through payment of an article-processing charge (APC). Authors will not be required to make a final decision about access to their article until it has been accepted. [Find out more about Transformative Journals](https://www.springernature.com/gp/open-research/transformative-journals)

Your paper will now be copyedited to ensure that it conforms to Nature Methods style. Once proofs are generated, they will be sent to you electronically and you will be asked to send a corrected version within 24 hours. It is extremely important that you let us know now whether you will be difficult to contact over the next month. If this is the case, we ask that you send us the contact information (email, phone and fax) of someone who will be able to check the proofs and deal with any last-minute problems.

If, when you receive your proof, you cannot meet the deadline, please inform us at rjsproduction@springernature.com immediately.

Once your manuscript is typeset and you have completed the appropriate grant of rights, you will receive a link to your electronic proof via email with a request to make any corrections within 48 hours. If, when you receive your proof, you cannot meet this deadline, please inform us at rjsproduction@springernature.com immediately.

Once your paper has been scheduled for online publication, the Nature press office will be in touch to confirm the details.

Once your paper has been scheduled for online publication, the Nature press office will be in touch to confirm the details.

Content is published online weekly on Mondays and Thursdays, and the embargo is set at 16:00 London time (GMT)/11:00 am US Eastern time (EST) on the day of publication. If you need to know the exact publication date or when the news embargo will be lifted, please contact our press office after you have submitted your proof corrections. Now is the time to inform your Public Relations or Press Office about your paper, as they might be interested in promoting its publication. This will allow them time to prepare an accurate and satisfactory press release. Include your manuscript tracking number NMETH-A45365C and the name of the journal, which they will need when they contact our office.

About one week before your paper is published online, we shall be distributing a press release to news organizations worldwide, which may include details of your work. We are happy for your institution or funding agency to prepare its own press release, but it must mention the embargo date and Nature Methods. Our Press Office will contact you closer to the time of publication, but if you or your Press Office have any inquiries in the meantime, please contact press@nature.com.

Nature Portfolio journals [encourage authors to share their step-by-step experimental protocols](https://www.nature.com/nature-research/editorial-policies/reporting-standards#protocols) on a protocol sharing platform of their choice. Nature Portfolio 's Protocol Exchange is a free-to-use and open resource for protocols; protocols deposited in Protocol Exchange are citable and can be linked from the published article. More details can found at www.nature.com/protocolexchange/about.

Please note that you and any of your coauthors will be able to order reprints and single copies of the issue containing your article through Nature Portfolio 's reprint website, which is located at <http://www.nature.com/reprints/author-reprints.html>. If there are any questions about reprints please send an email to author-reprints@nature.com and someone will assist you.

Best regards,
Allison

Allison Doerr, Ph.D.
Chief Editor
Nature Methods